# COUNTERFACTUAL GENERATIVE NETWORKS

**Axel Sauer**[1,2] **& Andreas Geiger**[1,2]
Autonomous Vision Group
[1]Max Planck Institute for Intelligent Systems, Tübingen    [2]University of Tübingen
{firstname.lastname}@tue.mpg.de

## ABSTRACT

Neural networks are prone to learning shortcuts – they often model simple correlations, ignoring more complex ones that potentially generalize better. Prior works on image classification show that instead of learning a connection to object shape, deep classifiers tend to exploit spurious correlations with low-level texture or the background for solving the classification task. In this work, we take a step towards more robust and interpretable classifiers that explicitly expose the task's causal structure. Building on current advances in deep generative modeling, we propose to decompose the image generation process into independent causal mechanisms that we train without direct supervision. By exploiting appropriate inductive biases, these mechanisms disentangle object shape, object texture, and background; hence, they allow for generating *counterfactual images*. We demonstrate the ability of our model to generate such images on MNIST and ImageNet. Further, we show that the counterfactual images can improve out-of-distribution robustness with a marginal drop in performance on the original classification task, despite being synthetic. Lastly, our generative model can be trained efficiently on a single GPU, exploiting common pre-trained models as inductive biases.

## 1 INTRODUCTION

Deep neural networks (DNNs) are the main building blocks of many state-of-the-art machine learning systems that address diverse tasks such as image classification (He et al., 2016), natural language processing (Brown et al., 2020), and autonomous driving (Ohn-Bar et al., 2020). Despite the considerable successes of DNNs, they still struggle in many situations, e.g., classifying images perturbed by an adversary (Szegedy et al., 2013), or failing to recognize known objects in unfamiliar contexts (Rosenfeld et al., 2018) or from unseen poses (Alcorn et al., 2019).

Many of these failures can be attributed to dataset biases (Torralba & Efros, 2011) or shortcut learning (Geirhos et al., 2020). The DNN learns the simplest correlations and tends to ignore more complex ones. This characteristic becomes problematic when the simple correlation is spurious, i.e., not present during inference. The motivational example of (Beery et al., 2018) considers the setting of a DNN that is trained to recognize cows in images. A real-world dataset will typically depict cows on green pastures in most images. The most straightforward correlation a classifier can learn to predict the label "cow" is hence the connection to a green, grass-textured background. Generally, this is not a problem during inference as long as the test data follows the same distribution. However, if we provide the classifier an image depicting a purple cow on the moon, the classifier should still confidently assign the label "cow." Thus, if we want to achieve robust generalization beyond the training data, we need to disentangle possibly spurious correlations from causal relationships.

Distinguishing between spurious and causal correlations is one of the core questions in causality research (Pearl, 2009; Peters et al., 2017; Schölkopf, 2019). One central concept in causality is the assumption of *independent mechanisms* (IM), which states that a causal generative process is composed of autonomous modules that do not influence each other. In the context of image classification (e.g., on ImageNet), we can interpret the generation of an image as a causal process (Kocaoglu et al., 2018; Goyal et al., 2019; Suter et al., 2019). We decompose this process into separate IMs, each controlling one factor of variation (FoV) of the image. Concretely, we consider three IMs: one generates the object's shape, the second generates the object's texture, and the third generates the background. With access to these IMs, we can produce *counterfactual images*, i.e., images of unseen combinations of FoVs. We can then train an ensemble of invariant classifiers on the generated coun-

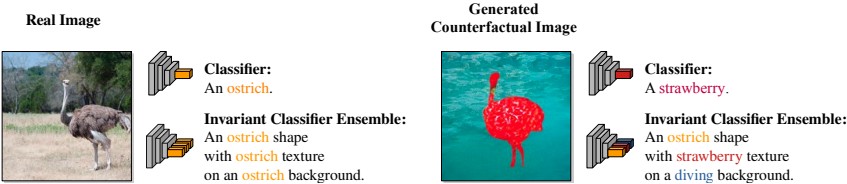

Figure 1: **Out-of-Domain (OOD) Classification.** A classifier focuses on all factors of variation (FoV) in an image. For OOD data, this can be problematic: a FoV might be a spurious correlation, hence, impairing the classifier's performance. An ensemble, e.g., a classifier with a common backbone and multiple heads, each head invariant to all but one FoV, increases OOD robustness.

terfactual images, such that every classifier relies on only a single one of those factors. The main idea is illustrated in Figure 1. By exploiting concepts from causality, this paper links two previously distinct domains: disentangled generative models and robust classification. This allows us to scale our experiments beyond small toy datasets typically used in either domain. The main contributions of our work are as follows:

- We present an approach for generating high-quality counterfactual images with direct control over shape, texture, and background. Supervision is only provided by the class label and certain inductive biases we impose on the learning problem.
- We demonstrate the usefulness of the generated counterfactual images for the downstream task of image classification on both MNIST and ImageNet. Our model improves the classifier's out-of-domain robustness while only marginally degrading its overall accuracy.
- We show that our generative model demonstrates interesting emerging properties, such as generating high-quality binary object masks and unsupervised image inpainting.

We release our code at https://github.com/autonomousvision/counterfactual_generative_networks

## 2 Structural Causal Models for Image Generation

In this section, we first introduce our ideas on a conceptual level. Concretely, we form a connection between the areas of causality, disentangled representation learning, and invariant classifiers, and highlight that domain randomization (Tobin et al., 2017) is a particular instance of these ideas. In section 3, we will then formulate a concrete model that implements these ideas for image classification. Our goals are two-fold: (i) We aim at generating counterfactual images with previously unseen combinations like a cat with elephant texture or the proverbial "bull in a china shop." (ii) We utilize these images to train a classifier invariant to chosen factors of variation.

In the following, we first formalize the problem setting we address. Second, we describe how we can address this setting by structuring a generator network as a structural causal model (SCM). Third, we show how to use the SCM for training robust classifiers.

### 2.1 Problem Setting

Consider a dataset comprised of (high-dimensional) observations $\mathbf{x}$ (e.g. images), and corresponding labels $y$ (e.g. classes). A common assumption is that each $\mathbf{x}$ can be described by lower-dimensional, semantically meaningful factors of variation $\mathbf{z}$ (e.g., color or shape of objects in the image). If we can *disentangle* these factors, we are able to control their influence on the classifier's decision. In the disentanglement literature, the factors are often assumed to be statistically independent, i.e., $\mathbf{z}$ is distributed according to $p(\mathbf{z}) = \Pi_{i=1}^{n}(z_i)$ (Locatello et al., 2018). However, assuming independence is problematic because certain factors might be correlated in the training data, or the combination of some factors may not exist. Consider the colored MNIST dataset (Kim et al., 2019), where both the digit's color and its shape correspond to the label. The simplest decision rule a classifier can learn is to count the number of pixels of a specific color value; no notion of the digit's shape is required. This kind of correlation is not limited to constructed datasets – classifiers trained on ImageNet (Deng et al., 2009) strongly rely on texture for classification, significantly more than on the object's shape (Geirhos et al., 2018). While texture or color is a powerful classification cue, we do not want the

classifier to ignore shape information completely. Therefore, we advocate a generative viewpoint. However, simply training, e.g., a disentangled VAE (Higgins et al., 2017) on this dataset, does not allow for generating data points of unseen combinations – the VAE cannot generate green zeros if all zeros in the training data are red (see Appendix A for a visualization). We therefore propose a novel generative model which enables full control over several FoVs relevant for classification. We then train a classifier on these images while randomizing all factors but one. The classifier focuses on the non-randomized factor and becomes invariant wrt. the randomized ones.

## 2.2 Structural Causal Models

In representation learning, it is commonly assumed that a potentially complex function $f$ generates images from a small set of high-level semantic variables (e.g., position or color of objects) (Bengio et al., 2013). Most previous work (Goyal et al., 2019; Suter et al., 2019) imposes no restrictions on $f$, i.e., a neural network is trained to map directly from a low-dimensional latent space to images. We follow the argument that rather than training a monolithic network to map from a latent space to images, the mapping should be decomposed into several functions. Each of these functions is autonomous, e.g., we can modify the background of an image while keeping all other aspects of the image unchanged. These demands coincide with the concept of structural causal models (SCMs) and independent mechanisms (IMs). An SCM $\mathfrak{C}$ is defined as a collection of $d$ (structural) assignments

$$S_j := f_j\left(\mathbf{PA}_j, U_j\right), \quad j = 1, \ldots, d \tag{1}$$

where each random variable $S_j$ is a function of its parents $\mathbf{PA}_j \subseteq \{S_1, \ldots, S_d\} \setminus \{S_j\}$ and a noise variable $U_j$. The noise variables $U_1, \ldots, U_d$ are jointly independent. The functions $f_i$ are independent mechanisms, intervening on one mechanism $f_j$ does not change the other mechanisms $\{f_1, \ldots, f_d\} \setminus \{f_j\}$. The SCM $\mathfrak{C}$ defines a unique distribution over the variables $\mathbf{S} = (S_1, \ldots, S_d)$ which is referred to as the *entailed distribution* $P_{\mathbf{S}}^{\mathfrak{C}}$. If one or more structural assignments are replaced, i.e., $S_k := \tilde{f}(\tilde{\mathbf{PA}}_k, \tilde{U}_k)$, this is called an intervention. We consider the case of *atomic interventions*, when $\tilde{f}(\tilde{\mathbf{PA}}_k, \tilde{U}_k)$ puts a point mass on a real value $a$. The entailed distribution then changes to the intervention distribution $P_{\mathbf{S}}^{\mathfrak{C};do(S_k:=a)}$, where the *do* refers to the intervention. A thorough review of these concepts can be found in (Peters et al., 2017). Our goal is to represent the image generation process with an SCM. If we learn a sensible set of IMs, we can intervene on a subset of them and generate *interventional images* $\mathbf{x}_{IV}$. These images were not part of the training data $\mathbf{x}$ as they are generated from the intervention distribution $P_{\mathbf{S}}^{\mathfrak{C};do(S_k:=a)}$. To generate a set of *counterfactual images* $\mathbf{x}_{CF}$, we fix the noise $u$ and randomly draw $a$, hence answering counterfactual questions such as "How would *this* image look like with a different background?". In our case, $a$ corresponds to a class label that we provide as input, denoted as $y_{CF}$ in the following.

## 2.3 Training an Invariant Classifier

To train an invariant classifier, we generate counterfactual images $\mathbf{x}_{CF}$, by intervening on all $f_j$ simultaneously. Towards this goal, we draw labels uniformly from the set of possible labels $\mathcal{Y}$ for each $f_j$, i.e., each IM is conditioned on a different label. We denote the domain of images generated by all possible label permutations as $\mathcal{X}_{\mathcal{CF}}$. The task of the invariant classifier $r : \mathcal{X}_{\mathcal{CF}} \to \mathcal{Y}_{CF,k}$ is then to predict the label $\mathbf{y}_{CF,k}$ that was provided to one specific IM $f_k$ – rendering $r$ invariant wrt. all other IMs. This type of invariance is reminiscent of the idea of domain randomization (Tobin et al., 2017). Here, the goal is to solve a robotics task while randomizing all task-irrelevant attributes. The randomization improves the performance of the learned policy in the real-world. In domain randomization, we commonly assume access to the true generative model (the simulator). This assumption is not feasible if we do not have access to this model. Similar connections of causality and data augmentation have been made in (Ilse et al., 2020). It is also possible to train on interventional images $\mathbf{x}_{IV}$, i.e., generating a single image per sampled noise vector. Empirically, we find that counterfactual images improve performance over interventional ones. We hypothesize that counterfactuals provide a more stable signal.

## 3 Counterfactual Generative Networks

In this section, we apply our ideas outlined above to the particular problem of image classification. Our goal is to decompose the image generation process into several IMs. In image classification,

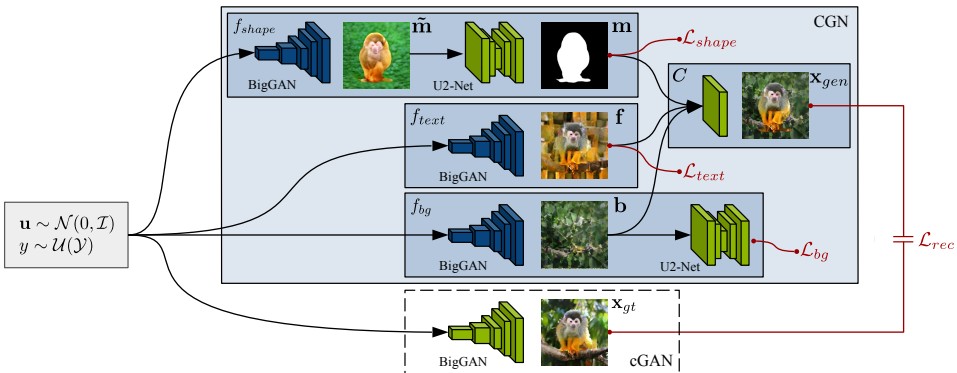

Figure 2: **Counterfactual Generative Network (CGN).** Here, we illustrate the architecture used for the ImageNet experiments. The CGN is split into four mechanisms, the shape mechanism $f_{shape}$, the texture mechanism $f_{text}$, the background mechanism $f_{bg}$, and the composer $C$. Components with trainable parameters are blue, components with fixed parameters are green. The primary supervision is provided by an unconstrained conditional GAN (cGAN) via the reconstruction loss $\mathcal{L}_{rec}$. The cGAN is only used for training, as indicated by the dotted lines. Each mechanism takes as input the noise vector $\mathbf{u}$ (sampled from a spherical Gaussian) and the label $y$ (drawn uniformly from the set of possible labels $\mathcal{Y}$) and minimizes its respective loss ($\mathcal{L}_{shape}$, $\mathcal{L}_{text}$, and $\mathcal{L}_{bg}$). To generate a set of counterfactual images, we sample $\mathbf{u}$ and then independently sample $y$ for each mechanism.

there is generally one principal object in the image. Hence, we assume three IMs for this specific task: object shape, object texture, and background. Our goal is to train the generator consisting of these mechanisms in an end-to-end manner. The inherent structure of the model allows us to generate meaningful counterfactuals by construction. In the following, we describe the inductive biases we use (network architectures, losses, pre-trained models) and how to train the invariant classifier. We refer to the entire generative model using IMs as a Counterfactual Generative Network (CGN).

## 3.1 INDEPENDENT MECHANISMS

We assume the causal structure to be known, and consider three learned IMs for generating shape, texture, and background, respectively. The only difference between the MNIST variants and ImageNet is the background mechanism. For the MNIST variants, we can simplify the SCM to include a second texture mechanism instead of a dedicated background mechanism. There is no need for a globally coherent background in the MNIST setting. An explicit formulation of both SCM is shown in Appendix B. In both cases, the learned IMs feed into another, fixed, IM: the composer. An overview of our CGN is shown in Figure 2. All IM-specific losses are optimized jointly end-to-end. For the experiments on ImageNet, we initialize each IM backbone with weights from a pre-trained BigGAN-deep-256 (Brock et al., 2018), the current state-of-the-art for conditional image generation. BigGAN has been trained as a single monolithic function; hence, it cannot generate images of only texture or only background, since these would be outside of the training domain.

**Composition Mechanism.** The function of the composer is not learned but defined analytically. For this work, we build on common assumptions from compositional image synthesis (Yang et al., 2017) and deploy a simple image formation model. Given the generated masks, textures and backgrounds, we composite the image $\mathbf{x}_{gen}$ using alpha blending, denoted as $C$:

$$\mathbf{x}_{gen} = C(\mathbf{m}, \mathbf{f}, \mathbf{b}) = \mathbf{m} \odot \mathbf{f} + (1 - \mathbf{m}) \odot \mathbf{b} \tag{2}$$

where $\mathbf{m}$ is the mask (or alpha map), $\mathbf{f}$ is the foreground, and $\mathbf{b}$ is the background. The operator $\odot$ denotes elementwise multiplication. While, in general, IMs may be stochastic (Eq. 1), we did not find this to be necessary for the composer; therefore, we leave this mechanism deterministic. This fixed composition is a strong inductive bias in itself – the generator needs to generate realistic images through this bottleneck. To optimize the composite image, we could use an adversarial loss between real and composite images. While applicable to simple datasets such as MNIST, we found that an adversarial approach does not scale well to more complex datasets like ImageNet. To get a stronger and more stable supervisory signal, we, therefore, use an unconstrained, conditional GAN

(cGAN) to generate pseudo-ground-truth images $\mathbf{x}_{gt}$ from noise $\mathbf{u}$ and label $y$. We feed the same $\mathbf{u}$ and $y$ into the IMs to generate $\mathbf{x}_{gen}$ and minimize a reconstruction loss $\mathcal{L}_{rec}(\mathbf{x}_{gt}, \mathbf{x}_{gen})$. We find a combination of L1 loss and perceptual loss (Johnson et al., 2016) to work well. Note that during training, we utilize the same noise $\mathbf{u}$ and label $y$ to reconstruct the image generated by the cGAN. However, at inference time, we generate counterfactual images by randomizing both $\mathbf{u}$ and $y$ separately per mechanism.

**Shape Mechanism.** We model the shape using a binary mask predicted by shape IM $f_{shape}$, where 0 corresponds to the background and 1 to the object. Effectively, this mechanism implements foreground segmentation. The loss is comprised of two terms: $\mathcal{L}_{binary}$ and $\mathcal{L}_{mask}$. $\mathcal{L}_{binary}$ is the pixel-wise binary entropy of the mask; hence, minimizing it forces the output to be close to either 0 or 1. $\mathcal{L}_{mask}$ prohibits trivial solutions, i.e., masks with all 0's or 1's that are outside of a defined interval (see Appendix C for details). As we utilize a BigGAN backbone for our ImageNet-Experiments, we need to extract a binary mask from the backbone's output. Therefore, we add a pre-trained U2-Net (Qin et al., 2020) as a head on top of the BigGAN backbone. The U2-Net was trained for salient object detection on DUTS-TR (10553 images) (Wang et al., 2017). Hence, it is class agnostic; it generates an object mask for a salient object in the image. While the U2-Net presents a strong bias towards binary object masks, it does not fully solve the task at hand as it captures non-class specific parts (e.g., parts of trees in an elephant-class picture, see Figure 5). By fine-tuning the BigGAN backbone, we learn to generate images of the relevant part with exaggerated features to increase saliency. We refer to these as pre-masks $\tilde{\mathbf{m}}$.

**Texture Mechanism.** The texture mechanism $f_{text}$ is responsible for generating the foreground object's appearance, while not capturing any object shape or background cues. For MNIST, we use an architectural bias – an additional layer before the final output. This layer spatially divides its input into patches and randomly rearranges them, similar to a shuffled sliding puzzle. This conceptually simple idea does not work on ImageNet, as we want to preserve local object structure, e.g., the position of an eye. We, therefore, sample patches from the full composite image and concatenate them into a grid. We denote this patch grid as $\mathbf{pg}$. The patches are sampled from regions where the mask values are highest (hence, the object is likely located). We then minimize a perceptual loss between the foreground $\mathbf{f}$ (the output of $f_{text}$) and the patchgrid: $\mathcal{L}_{text}(\mathbf{f}, \mathbf{pg})$. Over training, the background gradually transforms into object texture, resulting in texture maps, as shown in Figure 5. More details can be found in Appendix C.

**Background Mechanism.** The background mechanism $f_{bg}$ needs to capture the background's global structure while the object must be removed and inpainted realistically. However, we found that we cannot use standard inpainting techniques because classical methods (Barnes et al., 2009) slow down training too much, and deep learning methods (Liu et al., 2018) do not work well on synthetic data because of the domain shift. Instead, we exploit the same U2-Net as used for the shape mechanism $f_{shape}$. Again, we feed the output of the BigGAN backbone through the U2-Net with fixed weights. However, this time, we *minimize* the predicted saliency. Over the progress of training, this leads to the object shrinking and finally disappearing, while the model learns to inpaint the object region (see Figure 5 and Appendix E). We refer to this loss as $\mathcal{L}_{bg}$. We attribute this loss's success mainly the powerful pre-trained backbone network. BigGAN is already able to generate objects on realistic backgrounds; it only needs to unlearn the object generation.

## 3.2 GENERATING COUNTERFACTUALS TO TRAIN CLASSIFIERS

After training our CGN, each IM network has learned a class-conditional distribution over shapes, textures, or backgrounds. By randomizing the label input $y$ and noise $\mathbf{u}$ of each network, we can generate counterfactual images. The number of possible combinations is the number of classes to the power of the number of IM's. For ImageNet, this is $1000^3$. The amount of possible images is even larger since we learn distributions, i.e., we can generate a nearly unlimited variety of shapes, textures, and backgrounds, per class. We train on both real and counterfactual images. For MNIST, more counterfactual images always increase the test domain results; see the ablation study in Appendix A.3. On Imagenet, we provide evenly sized batches of real and counterfactual images; i.e., we use a ratio of 1. A ratio below 1 leads to inferior performance; a ratio above 1 leads to longer training times without an increase in performance. Similar results were reported for training on BigGAN samples in Ravuri & Vinyals (2019).

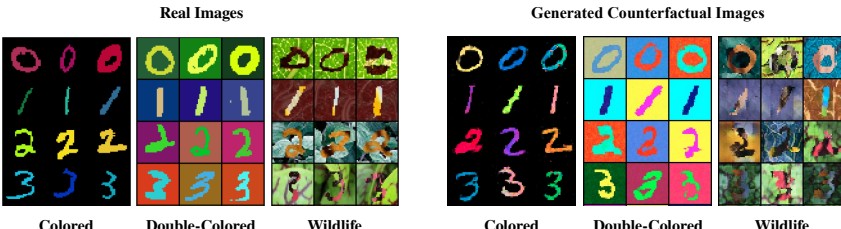

Figure 3: **MNISTs.** Left: Samples of the different MNIST variations (for brevity, we show only the first four classes). Right: Counterfactual samples generated by our CGN. Note that the CGN learned class-conditional *distributions*, i.e., it generates varying shapes, colors, and textures.

## 4 EXPERIMENTS

Our experiments aim to answer the following questions: (i) Does our approach reliably learn the disentangled IMs on datasets of different complexity? (ii) Which inductive biases are necessary to achieve this? (iii) Do counterfactual images enable training invariant classifiers? We first apply our approach to different versions of MNIST: colored-, double-colored- and Wildlife-MNIST (details about their generation are in Appendix A.2). The label is encoded in the digit shape, foreground color or texture, and the background color or texture, see Figure 3. Our work focuses on a setting where the spurious signal is a strong predictor of the label; hence we assume a correlation strength of at least 90 % between signal and label in our simulated environments. This assumption is in line with latest related work on visual bias (Goyal et al., 2019; Wang et al., 2020), which considers a strong correlation to be above 95 %. We then scale our approach to ImageNet and demonstrate that we can improve the robustness of ImageNet classifiers. Implementation details about architectures, loss parameters, and hyperparameters can be found in Appendix C.

### 4.1 DOES OUR APPROACH LEARN THE DISENTANGLED INDEPENDENT MECHANISMS?

Standard metrics like the Inception Score (IS) (Salimans et al., 2016) are not applicable since the counterfactual images are outside of the natural image domain. We thus focus on qualitative results in this section. For a quantitative analysis, we refer the reader to Section 4.3 where we analyze the accuracy, robustness, and invariance of classifiers trained on the generated counterfactual data.

**MNISTs.** The generated counterfactual images are shown in Figure 3 (right). None of the counterfactual combinations were present in the training data. We can see that CGN successfully generates high-quality counterfactuals. The results on Wildlife MNIST are surprisingly good, considering that the object texture is only observable on the relatively thin digits. Nevertheless, the texture IM learns to generate realistic textures. All experiments on MNIST are done without pre-training any network.

**ImageNet.** As shown in Figure 4, our CGN generates counterfactuals of high visual fidelity. We train a single CGN for all 1000 classes. We also find an unexpected benefit of our approach. In some instances, the composite images eliminate structural artifacts of the original BigGAN images, such as surplus legs, as shown in Figure 5. We hypothesize that $f_{shape}$ learns a general shape concept per class, resulting in outliers, like elephants with eight legs, being smoothed out. We show more samples, individual IM outputs, and interpolations in Appendix D. The CGN can fail to produce high-quality texture maps for very small objects, e.g., for a bird high up in the sky, the texture map will still show large portions of the sky. Also, in some instances, a residue of the object is left on the background, e.g., a dog snout. For generating counterfactual images, this is not a problem as a different object will cover the residue. Lastly, the enforced constraints can lead to a reduction in realism of the composite images $\mathbf{x_{gen}}$ compared to the original BigGAN samples. We show examples and propose solutions for these problems in Appendix F.

### 4.2 WHICH INDUCTIVE BIASES ARE NEEDED TO ACHIEVE DISENTANGLEMENT?

We employ two kinds of biases: pre-training of modules and IM-specific losses. We find that pre-training is not necessary for our experiments on MNIST. However, when scaling to ImageNet, powerful pre-trained models are key for achieving good results. Furthermore, this allows to train the

| Shape | red wine | cottontail rabbit | pirate ship | triumphal arch | mushroom | hyena dog |
|---|---|---|---|---|---|---|
| Texture | carbonara | head cabbage | banana | Indian elephant | barrel | school bus |
| Background | baseball | valley | bittern (bird) | viaduct | grey whale | snorkel |

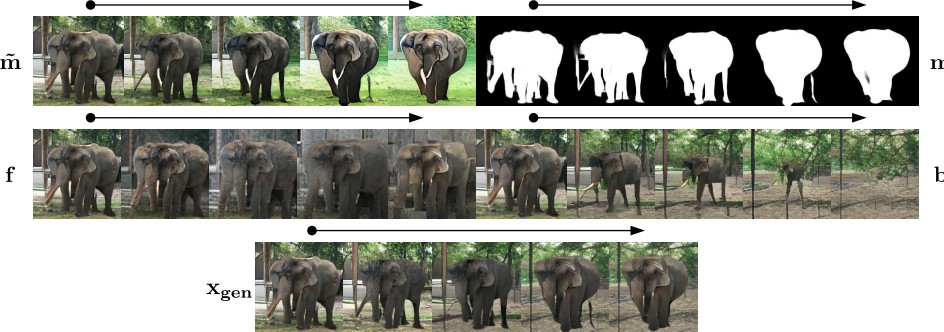

Figure 4: **ImageNet Counterfactuals.** The CGN successfully learns the disentangled shape, texture, and background mechanisms, and enables the generation of numerous permutations thereof.

Figure 5: **Individual IM Outputs over Training.** We show pre-masks $\tilde{\mathbf{m}}$, masks $\mathbf{m}$, foregrounds $\mathbf{f}$, and backgrounds $\mathbf{b}$. The arrows indicate the beginning and end of the training. The initial output of the pre-trained models is gradually transformed while the composite image only marginally changes.

whole CGN on a single NVIDIA GTX 1080Ti within 12 hours, in contrast to BigGAN, which was trained on a Google TPU v3 Pod with 512 cores for up to 48 hours. To investigate each loss' influence, we disable one loss at a time and measure its influence on the quality of the composite images. The composite images are on the image manifold, hence, we can calculate their Inception score (IS). As we train with pseudo ground truth, the performance of the unconstrained BigGAN is a natural upper bound. The used model reaches an IS of 202.9. To measure if the CGN collapsed during training, we monitor the mean value of the generated mask $\mu_{mask}$. A $\mu_{mask}$ close to 1 means that $f_{text}$ is not training. Instead, it generates the output of the pre-trained BigGAN, hence, a mask of 1's trivially minimizes the reconstruction loss $\mathcal{L}_{rec}(\mathbf{x}_{gt}, \mathbf{x}_{gen})$. The same is true for $\mu_{mask}$ close to 0 and $f_{bg}$. The results in Table 1 indicate that each loss is necessary, and jointly optimizing all of them end-to-end is needed for a high IS without a collapse of $\mu_{mask}$. Removing $\mathcal{L}_{shape}$, leads to bad quality masks (non-binary, only partially capturing the object). This results in a low IS since object texture and background get mixed in the composited image. Not using either $\mathcal{L}_{text}$ or $\mathcal{L}_{bg}$ results in a high IS (as the output is close to the original BigGAN output), but a collapse of $\mu_{mask}$. The mechanisms do not disentangle their respective signal. Finally, disabling $\mathcal{L}_{rec}$ leads to a very low IS, since the IMs can optimize their respective loss without any constraint on the composite image. We show the evolution and collapse of the masks over training in Appendix G.

### 4.3 Do counterfactual images enable Training of Invariant Classifiers?

The following experiments investigate if we can instill invariance into a classifier. We perform experiments on the MNIST variants, a cue-conflict dataset, and an OOD version of ImageNet.

**MNIST Classification.** In the training domain, shapes, colors, and textures are correlated with the class label. In the test domain, only the shapes correspond to the correct class. We compare to current approaches for training invariant classifiers: IRM (Arjovsky et al., 2019) and Learning-not-to-learn (LNTL) (Kim et al., 2019). For a detailed description we refer to Appendix C. *Original + CGN* is additionally trained on counterfactual data to predict the input labels of the shape IM. *Original + GAN* is a baseline that is trained on real and generated, non-counterfactual samples. IRM considers a signal to be causal if it is stable across several environments. We train IRM on 2 environments (90 % and 100 % correlation) or 5 environments (90 %, 92.5 %, 95 %, 97.5 %, and 100% correlation). LNTL considers color to be spurious, whereas we assume (complementary) that shapes are causal. Alternatively, we can follow the same assumption as IRM with an additional causal identification

| $\mathcal{L}_{shape}$ | $\mathcal{L}_{text}$ | $\mathcal{L}_{bg}$ | $\mathcal{L}_{rec}$ | IS ⇑ | $\mu_{mask}$ |
|:---:|:---:|:---:|:---:|:---:|:---:|
| ✗ | ✓ | ✓ | ✓ | 85.9 | 0.2 ± 0.2 % |
| ✓ | ✗ | ✓ | ✓ | 198.4 | 0.9 ± 0.1 % |
| ✓ | ✓ | ✗ | ✓ | 195.6 | 0.1 ± 0.1 % |
| ✓ | ✓ | ✓ | ✗ | 38.39 | 0.3 ± 0.2 % |
| ✓ | ✓ | ✓ | ✓ | 130.2 | 0.3 ± 0.2% |
| BigGAN (Upper Bound) | | | | 202.9 | - |

Table 1: **Loss Ablation Study.** We turn off one loss at a time. Values indicating mask collapse are red.

| | colored MNIST | | double-colored MNIST | | Wildlife MNIST | |
|---|---|---|---|---|---|---|
| | Train Acc ⇑ | Test Acc ⇑ | Train Acc ⇑ | Test Acc ⇑ | Train Acc ⇑ | Test Acc ⇑ |
| Original | 99.5 % | 35.9 % | 100.0 % | 10.3 % | 100.0 % | 10.1 % |
| IRM (2 Envs) | 99.6 % | 59.8 % | 100.0 % | 67.7 % | 99.9 % | 11.3 % |
| IRM (5 Envs) | - | - | 99.9 % | 78.9 % | 99.8 % | 76.8 % |
| LNTL | 99.3 % | 81.8 % | 98.7 % | 69.9 % | 99.9 % | 11.5 % |
| Original + GAN | 99.8 % | 40.7 % | 100.0 % | 10.8 % | 100.0 % | 10.4 % |
| Original + CGN | 99.7 % | **95.1** % | 97.4 % | **89.0** % | 99.2 % | **85.7** % |

Table 2: **MNISTs Classification.** In the test set, colors and textures are randomized, only the digit's shape corresponds to the class label. Random performance is at $10\%$.

| Trained on | Shape Bias | top-1 IN Acc ⇑ | top-5 IN Acc ⇑ |
|---|---|---|---|
| IN | 21.39 % | 76.13 % | 92.86 % |
| SIN | 81.37 % | 60.18 % | 82.62 % |
| IN + SIN | 34.65 % | 74.59 % | 90.03 % |
| IN + CGN/Shape | 54.82 % | | |
| IN + CGN/Text | 16.67 % | 73.98 % | 91.71 % |
| IN + CGN/Bg | 22.89 % | | |

Table 3: **Shape vs. Texture.** We can control the classifier's shape or texture preference.

| | Top-1 Test Accuracies | | | |
|---|---|---|---|---|
| Trained on | IN-9 ⇑ | Mixed-Same ⇑ | Mixed-Rand ⇑ | BG-Gap ⇓ |
| IN | 95.6% | 86.2% | 78.9% | 7.3% |
| SIN | 89.2 % | 73.1 % | 63.7 % | 9.4 % |
| IN + SIN | 94.7 % | 85.9 % | 78.5 % | 7.4 % |
| Mixed-Rand | 73.3% | 71.5% | 71.3% | 0.2 % |
| IN + CGN | 94.2 % | 83.4 % | 80.1 % | 3.3 % |

Table 4: **Accuracies on IN-9.** The reported accuracies are all obtained using a Resnet-50.

step, see Appendix H. The results in Table 2 confirm that training on counterfactual data leads to classifiers that are invariant to the spurious signals. We hypothesize that the difference between environments may be hard to pick up for IRM, especially if only a few are available. We find that we can further improve IRM's performance by adding more environments. However, continually increasing the number of environments is an unrealistic premise and only feasible in simulated environments. Our results indicate that LNTL and IRM have trouble scaling to more complex data.

**Texture vs. Shape Bias.** The Cue Conflict dataset consists of images generated using iterative style transfer (Gatys et al., 2015) between a texture and a content image. A high shape bias corresponds to classification according to the content label and vice versa for texture. Their approach is trained on stylized ImageNet (SIN), either as a drop-in for ImageNet (IN) or as augmentation. We use a classifier ensemble, i.e., a classifier with a common backbone and multiple heads, each head invariant to all but one FoV. We average the predicted log-probabilies of each head for the final output of the ensemble. We conduct all experiments using a Resnet-50 architecture. As shown in Table 3, we can influence the individual bias of each classifier head without significant degradation in the ensemble's performance.

**Invariance over Backgrounds.** Xiao et al. (2020) propose the *BG-Gap* to measure a classifier's dependence on the background signal. Based on ImageNet-9 (IN-9), a subset of ImageNet with 9 coarse-grained classes, they build synthetic datasets. For *Mixed-Rand*, the backgrounds are randomized, while the object remains unchanged, hence background an class are decorrelated. For *Mixed-Same* they sample class-consistent backgrounds. The BG-Gap is the difference in performance between the two. Training on IN or SIN does not make it possible to disentangle and omit the background signal, as shown in Table 4. Directly training on Mixed-Rand leads to a drop in performance on the original data which might be due to the smaller training dataset. We can generate unlimited data of this type, hence, we are able to reduce the gap while achieving high accuracy on IN-9. However, a gap to a fully invariant classifier remains. We partially attribute this to the general remaining domain gap between generated and real images.

## 5 RELATED WORK

Our work is related to disentangled representation learning and the training of invariant classifiers.

**Disentangled Representation Learning.** A recent line of work in image synthesis aims to learn disentangled features for controlling the image generation process (Chen et al., 2016; Higgins et al., 2017; Liao et al., 2020). The challenge of the task is that the underlying factors can be highly correlated. Closely related to our work is (Li et al., 2020), which aims to disentangle background, shape, pose, and texture, using object bounding boxes for supervision. Their methods assumes

images of a single object category (e.g. birds). We scale our approach to all classes of ImageNet which enables us to generate inter-class counterfactuals. A recent research direction explores the discovery of interpretable directions in GANs trained on ImageNet (Plumerault et al., 2020; Voynov & Babenko, 2020; Peebles et al., 2020). These approaches do not allow for generating counterfactual images. Kocaoglu et al. (2018) train two separate generative models, one generating binary feature labels (mustache, young), the other generating images conditioned on these labels. Their model can create images of previously unseen combinations of attributes, e.g., women with mustaches. This approach assumes a data set with fine-grained labels; hence it would not be suited to our application since labels for high-level concepts like shape are hard to obtain. Besserve et al. (2019) also leverage the idea of independent mechanisms to discover modularity in pre-trained generative models. Their approach does not allow for direct control of image attributes. Lastly, methods for causal generative modeling utilizing competing experts (von Kügelgen et al., 2020) have been demonstrated on toy datasets only. Further, none of the works above aim to use the generated images to improve upon a downstream task such as image classification.

**Invariant Classification.** Current approaches do not take an underlying causal model into account. Instead, they rely on different assumptions. Arjovsky et al. (2019) assume that the training data is collected into separate environments (e.g. different measurement circumstances). Correlations that are stable across environments are considered to be causal. Kim et al. (2019) aim to learn features that are uninformative of a given bias (spurious) signal. As mentioned above, attaining labels for shape or texture is expensive and not straight-forward. A recent strand of work is concerned with data augmentation for improving invariance against spurious correlations. Shetty et al. (2020) propose to train object detectors on generated semantic adversarial data, effectively reducing the texture dependency of their model. Their finding is in line with (Geirhos et al., 2018) that proposes to transfer the style of paintings onto images and use them for data augmentation. These approaches, however, do not allow to choose the specific signal we want invariance for, e.g., the background. The use of counterfactual data has been previously explored in natural language inference (Kaushik et al., 2020) and visual question answering (Teney et al., 2020).

## 6 DISCUSSION

We assume that an image can be neatly distinguished into a class foreground and background throughout this work. This assumption breaks once we consider more complex scenes with different object instances or for tasks without a clear foreground-background distinction, e.g., in medical images. The composition mechanism is a powerful bias, and crucial to making our model work. In other domains, equally strong biases may need to be identified to enable learning the SCM. An exciting research direction is to explore different configurations of IMs to tackle these challenges.

The additional constraints that we enforce during the CGN training lead to a reduced realism, as evidenced by the lower IS. We also find that our generated images can significantly influence a classifier's preference, but their quality is not high enough to *improve* performance on ImageNet. However, even state-of-the-art generative models (with higher IS) are not good enough yet to generate data for training competitive ImageNet classifiers (Ravuri & Vinyals, 2019).

Lastly, in our experiments, we assume the causal structure to be known. This assumption is substantially stronger than the ones in more general standard disentanglement frameworks (Chen et al., 2016; Higgins et al., 2017). A possible extension to our work could leverage causal discovery to isolate IMs in a domain-agnostic manner, e.g., via meta-learning (Bengio et al., 2020). On the other hand, the definition of a causal structure and the approximation through IMs may be a principled way to integrate domain knowledge into a machine learning system, The need for better interfaces to integrate domain knowledge has recently been highlighted in (D'Amour et al., 2020).

## 7 CONCLUSION

In this work, we apply ideas from causality to generative modeling and the training of invariant classifiers. We structure a generative network into independent mechanisms to generate counterfactual images useful for training classifiers. With the use of several inductive biases, we demonstrate our approach on various MNIST variants as well as ImageNet. Our ideas are orthogonal to advances in generative modeling - with advances therein, our obtained results will further improve.

ACKNOWLEDGMENTS

We acknowledge the financial support by the BMWi in the project KI Delta Learning (project number 19A19013O). Andreas Geiger was supported by the ERC Starting Grant LEGO-3D (850533). We would like to thank Yiyi Lao, Michael Niemeyer, and Elie Aljalbout for comments on an earlier paper draft and Songyou Peng, Michael Oechsle, and Kashyap Chitta for last-minute proofreading. We would also like to thank Vanessa Sauer for her general support and constructive criticism on the generated counterfactuals in earlier stages.

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

## APPENDIX A    MNIST VARIANTS

### A.1    VARIATIONAL AUTOENCODERS ON COLORED MNIST

Figure 6 shows the latent space of a $\beta$-VAE (Higgins et al., 2017) trained on colored MNIST. The VAE disentangles the data into different clusters present in the data. However, the axes do not correspond to color and shape, i.e., color *and* shape vary when traversing one latent. We used only two latent dimensions for visualization purposes; the problem is not resolved by adding more dimensions. The same behaviour can be observed for unconstrained GANs (Goodfellow et al., 2014), as a GAN also approximates the training distribution.

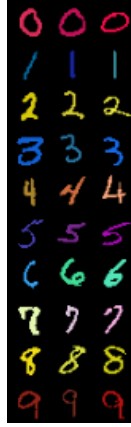 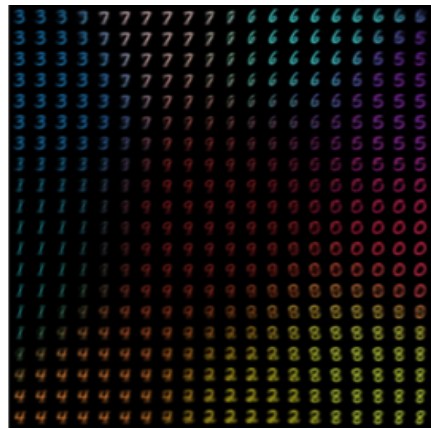

Figure 6: **Colored MNIST.** (left) Examples of data points. (right) Training a disentangled VAE with two latent dimensions on colored MNIST.

### A.2    MNISTs GENERATION

**Colored MNIST.** This dataset was proposed by Kim et al. (2019). They select ten distinct colors and assign each of them to a class. For each training image, they sample a color from a normal distribution with the class color as the mean. In the test set, the colors are randomly assigned. The variance $\sigma$ of the normal distribution can be used to control the amount of bias. We evaluate on the hardest, i.e., most biased, setting with $\sigma = 0.02$.

**Double-Colored MNIST.** We follow the same procedure as for colored MNIST. We additionally encode the class label in the background color.

**Wildlife MNIST.** To build a version of MNIST closer to an ImageNet setting, we add a texture bias to the data. We follow the same procedure as for double-colored MNIST. We take textures from (Cimpoi et al., 2014) and use ten images of the texture class "striped" to encode the label in the foreground. Similarly, we encode the label in the background with textures of the texture class "veiny." We do not add noise to texture to add stochasticity. Instead, we sample a 32x32 patch from the larger texture image. The textures are multi-modal; hence, these patches can look quite different.

### A.3    ABLATION STUDIES

In Figure 7, we study the effects of the number of counterfactual data points on the test accuracy. We increase the amount of counterfactual data while the amount of real data is fixed (50k for MNIST). We also ablate the amount of counterfactual drawn per sampled noise $u$. We find that the higher the number of counterfactual data points, the better. Also, it is advantageous to draw several counterfactuals per $u$. Our intuition is that several counterfactuals provide a more stable signal of the non-spurious factor.

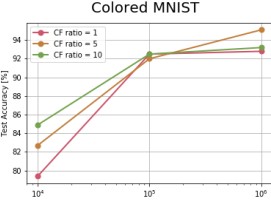 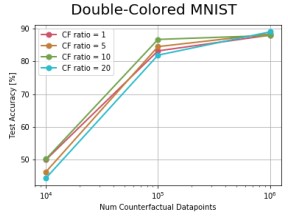 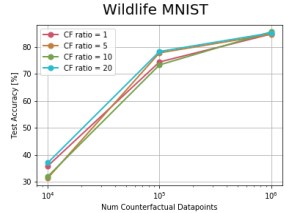

Figure 7: **MNIST Ablation Study.** To improve visibility, we start with $10^4$ counterfactual data points, below the performance is marginally better than the fully biased baseline. The CF ratio indicates how many counterfactuals we generate per sampled noise. For colored MNIST, the maximum CF ratio is ten as there are only ten possible colors per shape.

## APPENDIX B   CAUSAL STRUCTURES

The two SCM's are as follows:

<table>
<tr><th>MNISTs</th><th>ImageNet</th></tr>
<tr><td>$\mathbf{M} := f_{shape}(Y_1, U_1)$</td><td>$\mathbf{M} := f_{shape}(Y_1, U_1)$</td></tr>
<tr><td>$\mathbf{F} := f_{text,1}(Y_2, U_2)$</td><td>$\mathbf{F} := f_{text}(Y_2, U_2)$</td></tr>
<tr><td>$\mathbf{B} := f_{text,2}(Y_3, U_3)$</td><td>$\mathbf{B} := f_{bg}(Y_3, U_3)$</td></tr>
<tr><td>$\mathbf{X_{gen}} := C(\mathbf{M}, \mathbf{F}, \mathbf{B})$</td><td>$\mathbf{X_{gen}} := C(\mathbf{M}, \mathbf{F}, \mathbf{B})$</td></tr>
</table>

where $\mathbf{M}$ is the mask, $\mathbf{F}$ is the foreground, $\mathbf{B}$ is the background, $U_j$ is the exogenous noise, $Y_j$ is the class label, $\mathbf{X_{gen}}$ is the generated image, and $f_j$ and $C$ are the independent mechanisms.

## APPENDIX C   IMPLEMENTATION DETAILS

### C.1   SHAPE LOSS DETAILS

The full shape loss is as follows:

$$
\begin{aligned}
\mathcal{L}_{shape}(f_s) &= \mathcal{L}_{binary}(f_s) + \mathcal{L}_{mask}(f_s) \\
&= \mathbb{E}_{p(\mathbf{u},y)} \left[ \sum_{i=1}^{N} -m_i \log_2(m_i) - (1-m_i) * \log_2(1-m_i) \right] \\
&\quad + \mathbb{E}_{p(\mathbf{u},y)} \left[ \max\left(0, \tau - \frac{1}{N}\sum_{i=1}^{N} m_i\right) + \max\left(0, \frac{1}{N}\sum_{i=1}^{N} m_i - \tau\right) \right]
\end{aligned}
\tag{3}
$$

where $\mathcal{L}_{binary}$ is the pixel-wise binary entropy, $\mathcal{L}_{mask}$ is the mask loss, $\mathbf{m} = f_{shape}(\mathbf{u}, y)$ is the mask generated by $f_{shape}$ and $\tau$ is a scalar threshold. $\mathcal{L}_{binary}$ enforces the output to be close to either 0 or 1. $\mathcal{L}_{mask}$ prohibits trivial solutions, i.e., masks with all 0's or 1's, that are outside the interval defined by $\tau$. We set $\tau = 0.1$ in all experiments. A $\tau$ of 0.1 means that $\mu_{mask}$ is forced to be in the interval of $[0.1, 0.9]$ – the main object should occupy more than 10% and less than 90% of the image.

### C.2   TEXTURE LOSS DETAILS

The sampling procedure for $\mathcal{L}_{text}$ is as follows: we sample 36 patches of size $15 \times 15$ from the regions where the mask is closest to 1. Out of these 36 patches, we build a $6 \times 6$ patch grid. Finally, we upscale the grid to the full $256 \times 256$ resolution; we denote this grid as $\mathbf{pg}$. We then minimize a perceptual loss between the foreground $\mathbf{f}$ (the output of $f_{text}$) and the patchgrid: $\mathcal{L}_{text}(\mathbf{f}, \mathbf{pg})$.

### C.3 CGN TRAINING ON IMAGENET.

**Training Settings.** For training the CGN, we jointly optimize the following loss:

$$
\begin{aligned}
\mathcal{L} &= \mathcal{L}_{rec} + \mathcal{L}_{shape} + \lambda_5 \mathcal{L}_{text} + \lambda_6 \mathcal{L}_{bg} \\
&= \lambda_1 \mathcal{L}_{L1} + \lambda_2 \mathcal{L}_{perc} + \lambda_3 \mathcal{L}_{binary} + \lambda_4 \mathcal{L}_{mask} + \lambda_5 \mathcal{L}_{text} + \lambda_6 \mathcal{L}_{bg}
\end{aligned}
\tag{4}
$$

We use the following lambdas: $\lambda_1 = 100, \lambda_2 = 5, \lambda_3 = 300, \lambda_4 = 500, \lambda_5 = 5, \lambda_6 = 2000$. For the optimization we use Adam (Kingma & Ba, 2014), and set the learning rate of $f_{shape}$ to 8e-6, and for both $f_{text}$ and $f_{bg}$ to 1e-5. We do not use real data for training the CGN so we do not need to any data loading. We sample a single image from BigGAN and the CGN and accumulate the loss gradients for 4000 steps before taking a gradient step. Accumulating large pseudo-batches proved crucial for high-quality gradients, confirming the observations of (Brock et al., 2018). Further, using a batch size of one makes it possible to train on a single GPU. For the perceptual losses, i.e., $\mathcal{L}_{perc}$ and $\mathcal{L}_{text}$, we use pre-trained VGG16 with batch normalization layers. We calculate the style reconstruction loss (Johnson et al., 2016) of the features after the first four max-pooling layers.

We use the pre-trained BigGAN models from `https://github.com/huggingface/pytorch-pretrained-BigGAN`. We experiment with different values for the truncation value of the truncated normal distribution used to sample the input noise. However, we find it does not impact the performance of the CGN. Also, a low value leads to worse performance of the trained classifiers; hence, we leave the truncation value at 1.

**Hyperparameter Search.** We measure the Inception score (IS) during training and the mean value of the masks $\mu_{mask}$ to detect mask collapse. We also observe the generated images for a fixed noise vector; see the outputs in Figure 5. Our overall objective is a high IS and a stable $\mu_{mask}$. Further, we aim for high-quality output of all IMs (Masks: binary, capture only class-specific parts; Textures: no background/global shape visible, Background: no trace of foreground objects visible). We observe these outputs for several classes during optimization. The hyperparameters can be tuned mostly independently from each other, i.e., a better lambda for the mask loss does not influence the quality of the texture maps much.

### C.4 CLASSIFIER TRAINING

**MNIST**. We use the same CNN architecture for all experiments and approaches. For IRM, we produce versions of double-colored MNIST and Wildlife MNIST with different degree of correlation between the label and the foreground colors/textures and background colors/textures. We then train IRM on 2 environments (90% and 100% correlation) or 5 environments (90%, 92.5%, 95%, 97.5%, and 100% correlation). We schedule the gradient norm penalty weight, starting from 0, then linearly increasing it over the training episodes. We find the scheduling to be crucial for IRM to converge and achieve good performance.

**ImageNet**. We use a ResNet-50 from PyTorch `torchvision`. We share the weight up to the last layer as a common backbone and add three fully-connected heads (shape, texture, background). Each of the heads is provided its respective label when training on the counterfactual images. On the real images, we average the logits of all heads. This approach allows the single classifiers to focus on its assigned FoV while the ensemble performs well overall. Similarly to the experiments by (Geirhos et al., 2018), we begin the training with pre-trained weights from ImageNet. We train for 70 episodes with Stochastic Gradient Descent using a batch size of 512. Of the 512 images, 256 are real images, 256 are counterfactual images. We find that an even ratio between real and counterfactual images leads to the best results in terms of optimization stability and performance. We use a momentum of 0.9, weight decay (1e-4), and a learning rate of 0.1, multiplied by a factor of 0.001 after 30 and 60 epochs.

## APPENDIX D    MORE SAMPLES AND INTERPOLATIONS

### D.1    INDIVIDUAL IM OUTPUTS FOR DIFFERENT CLASS TYPES

In the following we illustrate the individual outputs of each IM for different classes. In each figure, we show from top to bottom: pre-masks $\tilde{\mathbf{m}}$, masks $\mathbf{m}$, texture maps $\mathbf{f}$, backgrounds $\mathbf{b}$, and composite images $\mathbf{x}_{gen}$. For all shown outputs, we set the truncation parameter for the noise to 0.5.

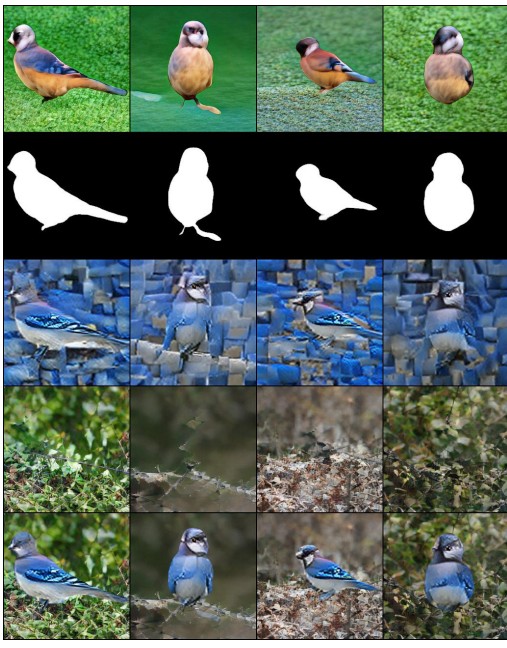

Figure 8: **IM Outputs for 'jay'.** From top to bottom: $\tilde{\mathbf{m}}, \mathbf{m}, \mathbf{f}, \mathbf{b}, \mathbf{x}_{gen}$.

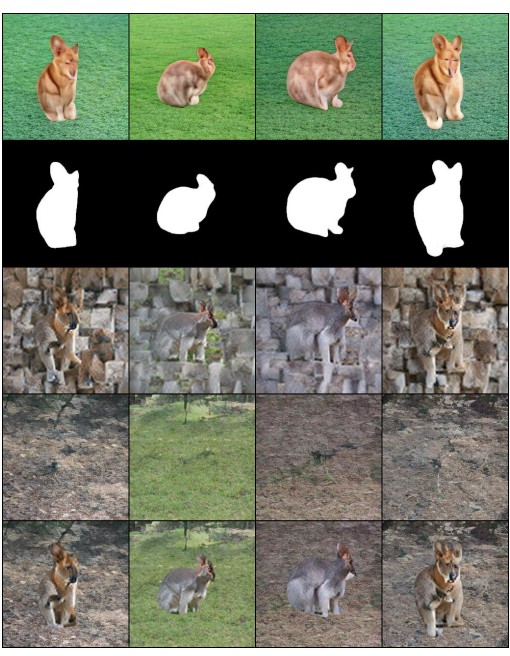

Figure 9: **IM Outputs for 'wallaby'.** From top to bottom: $\tilde{\mathbf{m}}, \mathbf{m}, \mathbf{f}, \mathbf{b}, \mathbf{x}_{gen}$.

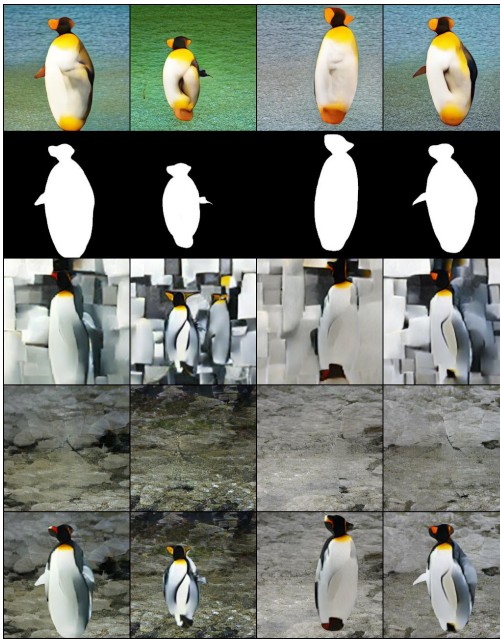

Figure 10: **IM Outputs for 'king penguin'.** From top to bottom: $\tilde{\mathbf{m}}, \mathbf{m}, \mathbf{f}, \mathbf{b}, \mathbf{x}_{gen}$.

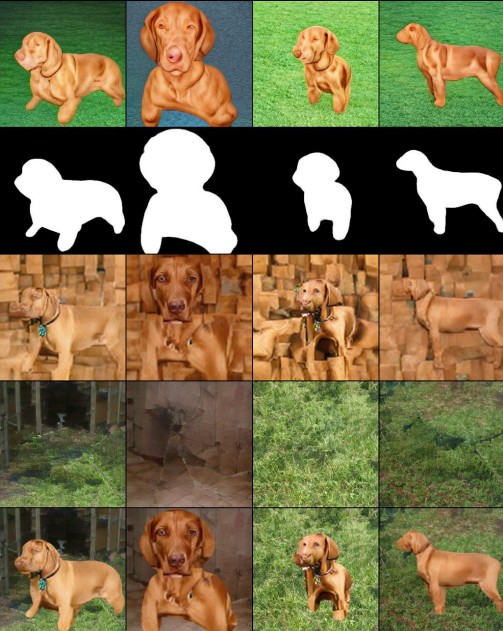

Figure 11: **IM Outputs for 'vizsla'.** From top to bottom: $\tilde{\mathbf{m}}, \mathbf{m}, \mathbf{f}, \mathbf{b}, \mathbf{x}_{gen}$.

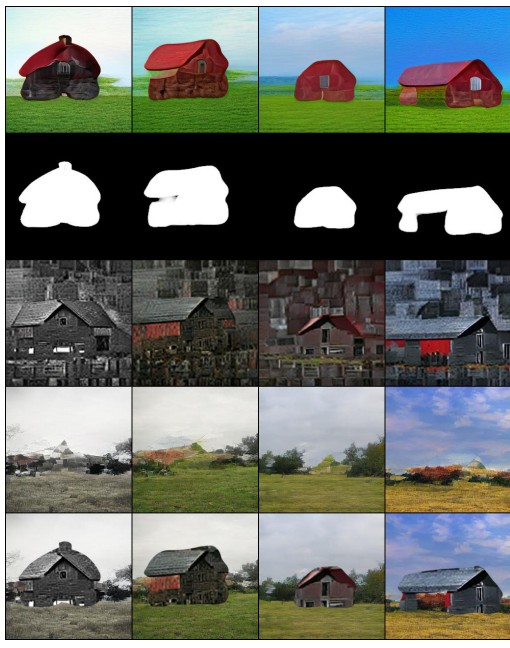

Figure 12: **IM Outputs for 'barn'**. From top to bottom: $\tilde{\mathbf{m}}, \mathbf{m}, \mathbf{f}, \mathbf{b}, \mathbf{x}_{gen}$.

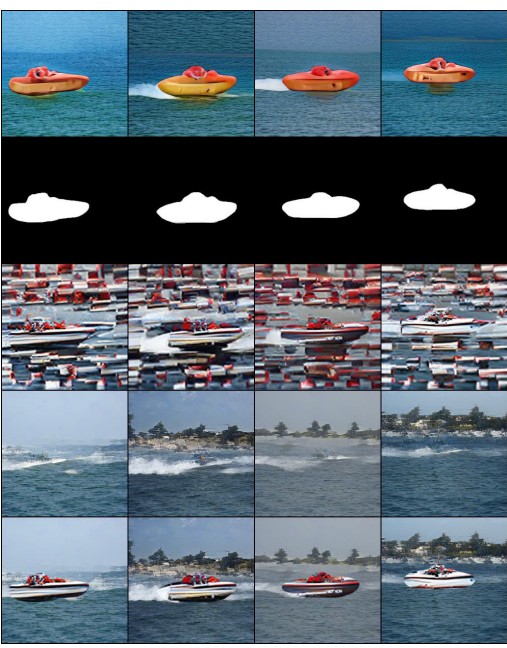

Figure 13: **IM Outputs for 'speedboat'.** From top to bottom: $\tilde{\mathbf{m}}, \mathbf{m}, \mathbf{f}, \mathbf{b}, \mathbf{x}_{gen}$.

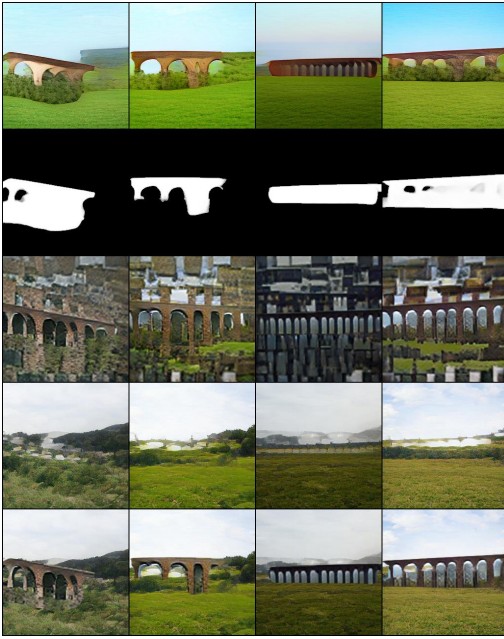

Figure 14: **IM Outputs for 'viaduct'**. From top to bottom: $\tilde{\mathbf{m}}, \mathbf{m}, \mathbf{f}, \mathbf{b}, \mathbf{x}_{gen}$.

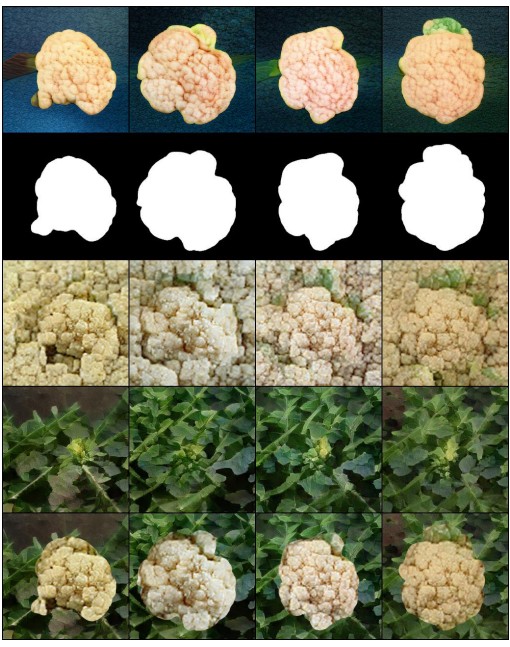

Figure 15: **IM Outputs for 'cauliflower'.** From top to bottom: $\tilde{\mathbf{m}}, \mathbf{m}, \mathbf{f}, \mathbf{b}, \mathbf{x}_{gen}$.

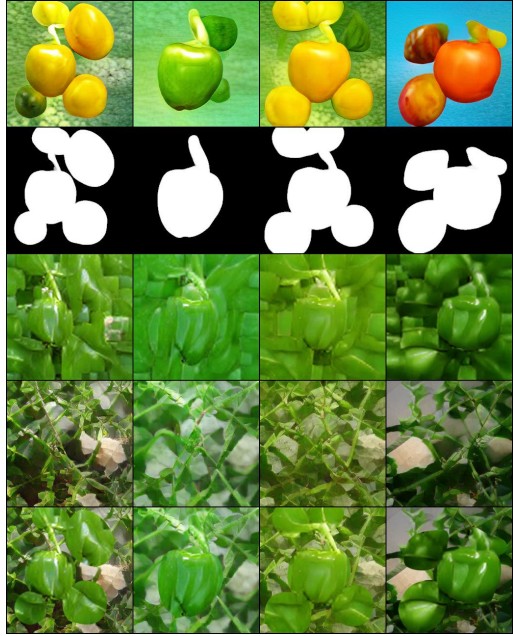

Figure 16: **IM Outputs for 'bell pepper'**. From top to bottom: $\tilde{\mathbf{m}}, \mathbf{m}, \mathbf{f}, \mathbf{b}, \mathbf{x}_{gen}$.

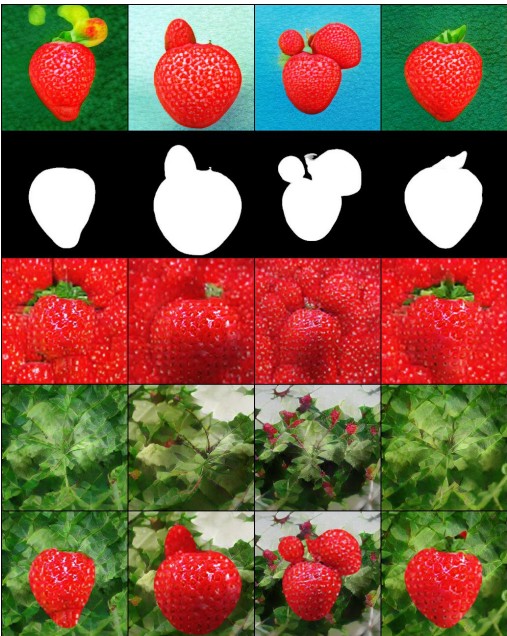

Figure 17: **IM Outputs for 'strawberry'.** From top to bottom: $\tilde{\mathbf{m}}, \mathbf{m}, \mathbf{f}, \mathbf{b}, \mathbf{x}_{gen}$.

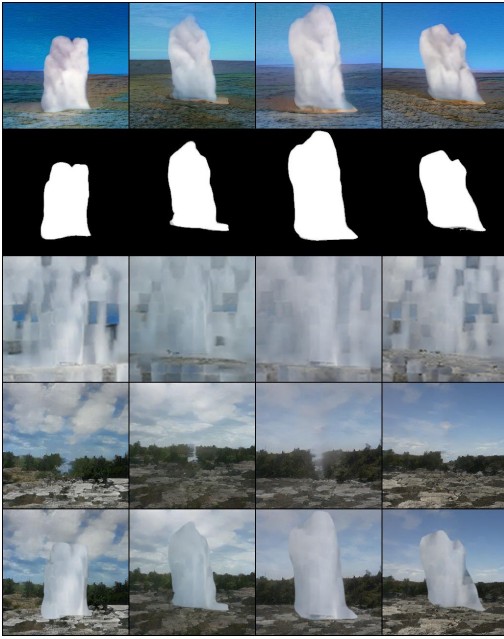

Figure 18: **IM Outputs for 'geyser'**. From top to bottom: $\tilde{\mathbf{m}}, \mathbf{m}, \mathbf{f}, \mathbf{b}, \mathbf{x}_{gen}$.

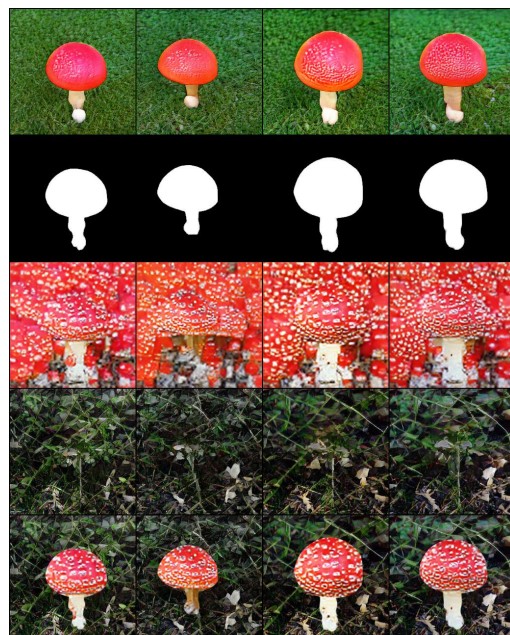

Figure 19: **IM Outputs for 'agaric'.** From top to bottom: $\tilde{\mathbf{m}}, \mathbf{m}, \mathbf{f}, \mathbf{b}, \mathbf{x}_{gen}$.

## D.2 LATENT INTERPOLATIONS

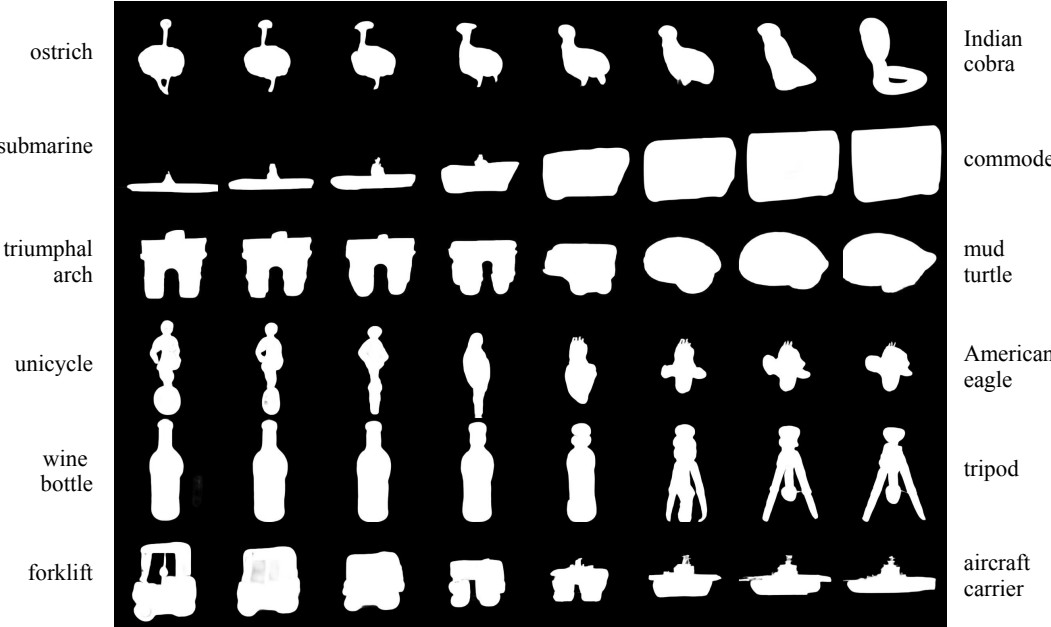

Figure 20: **Interpolating Shapes.** We interpolate between $u$ and $y$ pairs.

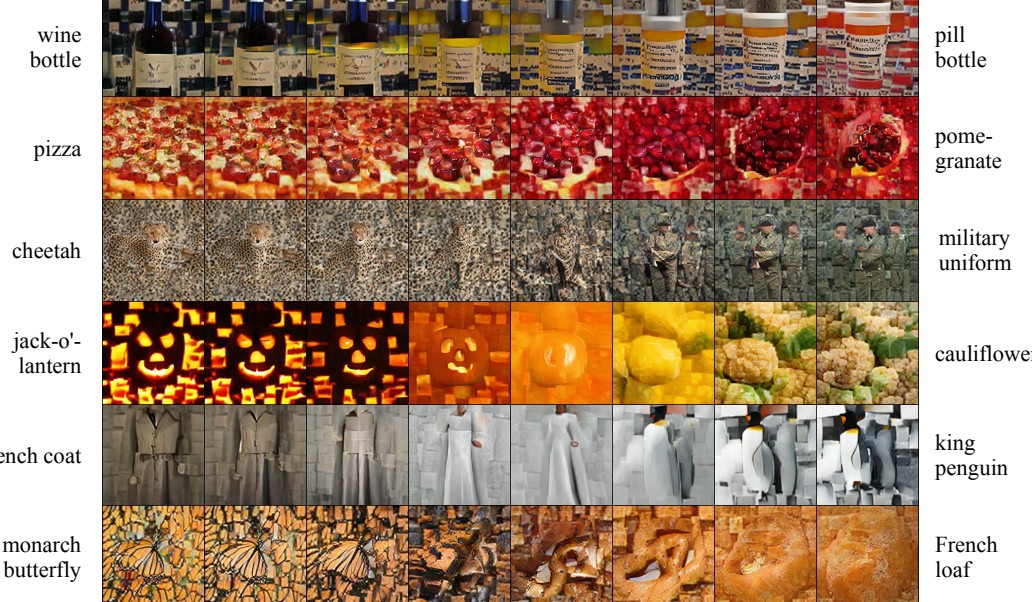

Figure 21: **Interpolating Textures.** We interpolate between $u$ and $y$ pairs.

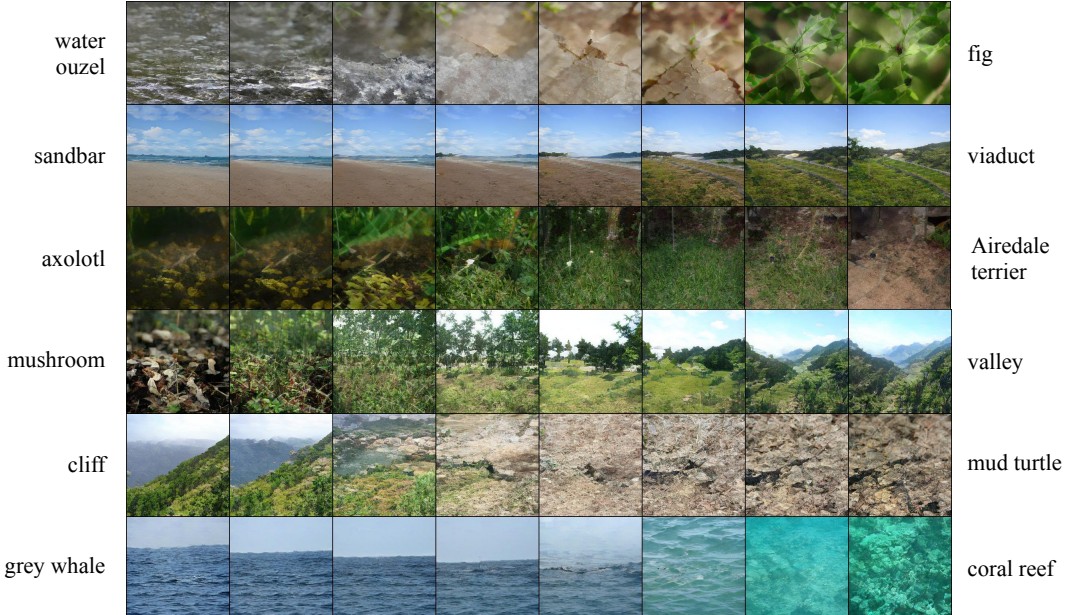

Figure 22: **Interpolating Backgrounds.** We interpolate between $u$ and $y$ pairs.

### D.3    MORE COUNTERFACTUAL SAMPLES

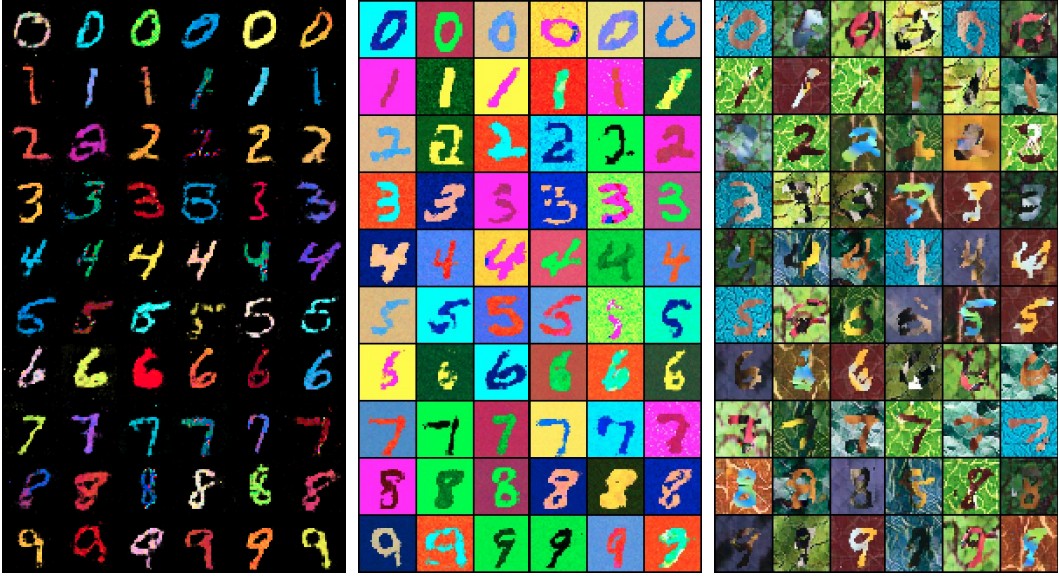

Figure 23: **MNIST Counterfactuals.** From Left to Right: colored, double-colored-, and Wildlife MNIST.

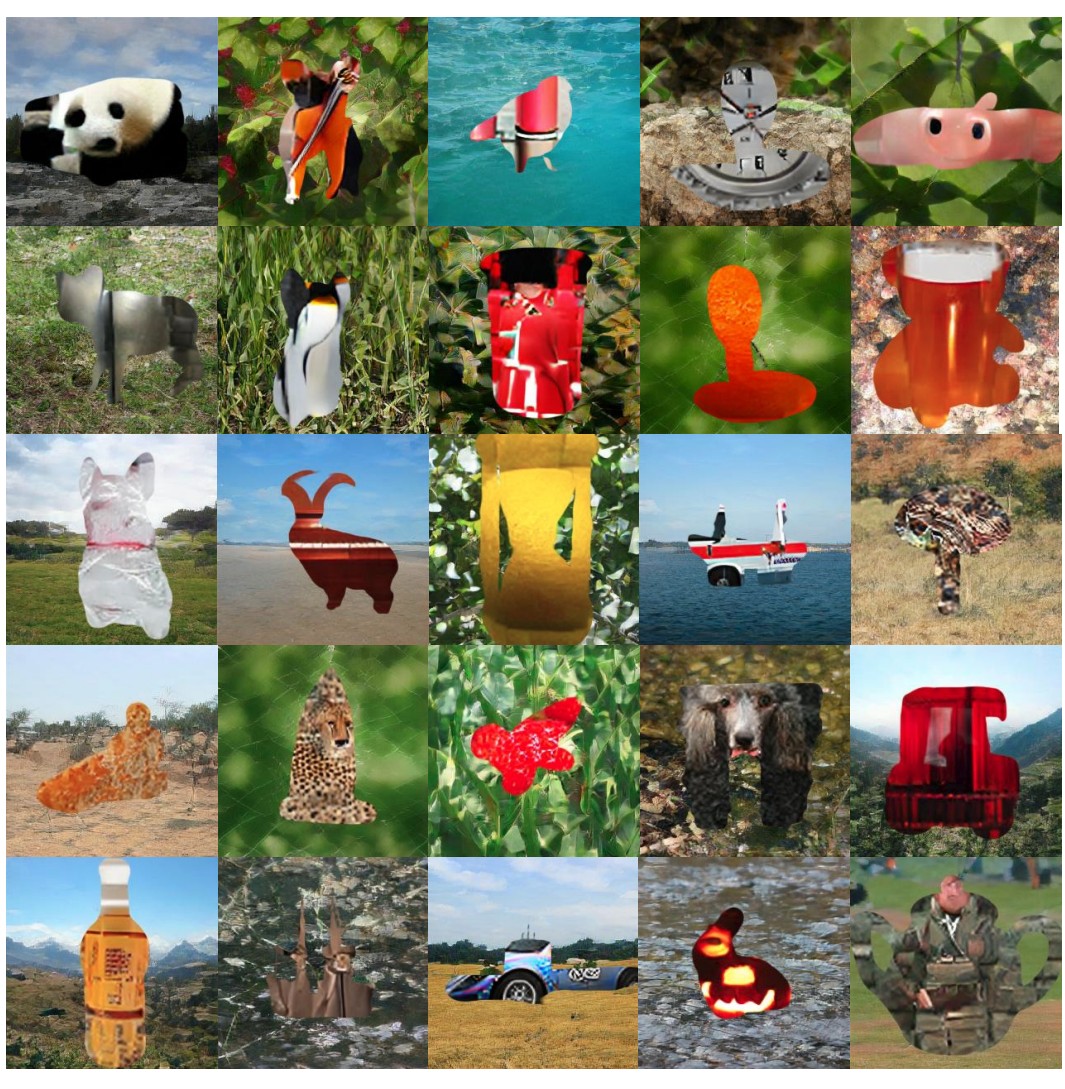

| Row | Column | Shape | Texture | Background | Row | Column | Shape | Texture | Background |
|-----|--------|-------|---------|------------|-----|--------|-------|---------|------------|
| 1 | 1 | school bus | panda | geyser | 3 | 4 | offshore rig | ambulance | breakwater |
| 1 | 2 | hyena dog | chello | strawberry | 3 | 5 | mushroom | sand viper | ostrich |
| 1 | 3 | robin | lipstick | diving | 4 | 1 | snowmobile | French Loaf | Arabian camel |
| 1 | 4 | Indian cobra | analog clock | green lizard | 4 | 2 | katamaran | cheetah | garden spider |
| 1 | 5 | race car | piggy bank | fig | 4 | 3 | bald eagle | strawberry | spike |
| 2 | 1 | hyena dog | breastplate | cock | 4 | 4 | triumphal arc | standard poodle | bullfrog |
| 2 | 2 | German shepherd | king penguin | bittern | 4 | 5 | forklift | theater curtain | valley |
| 2 | 3 | beaker | busby | pineapple | 5 | 1 | whine bottle | pill bottle | alp |
| 2 | 4 | Indian Cobra | orange | spider web | 5 | 2 | pirate ship | trench coat | beaver |
| 2 | 5 | teddy | beer glass | rock crab | 5 | 3 | submarine | race car | hay |
| 3 | 1 | malinois | reel | viaduct | 5 | 4 | wood rabbit | jack-o'-lantern | water ouzel |
| 3 | 2 | ibex | upright piano | sand bank | 5 | 5 | teapot | military uniform | baseball |
| 3 | 3 | hourglass | lemon | hornbill | | | | | |

Figure 24: **ImageNet Counterfactuals.** Top: Counterfactual Images. Bottom: ImageNet labels.

## APPENDIX E    IM OUTPUTS OVER THE COURSE OF TRAINING

In the following we illustrate the individual outputs of each IM over the course of training. In each figure, we show from top to bottom: pre-masks $\tilde{\mathbf{m}}$, masks $\mathbf{m}$, texture maps $\mathbf{f}$, backgrounds $\mathbf{b}$, and composite images $\mathbf{x}_{gen}$.

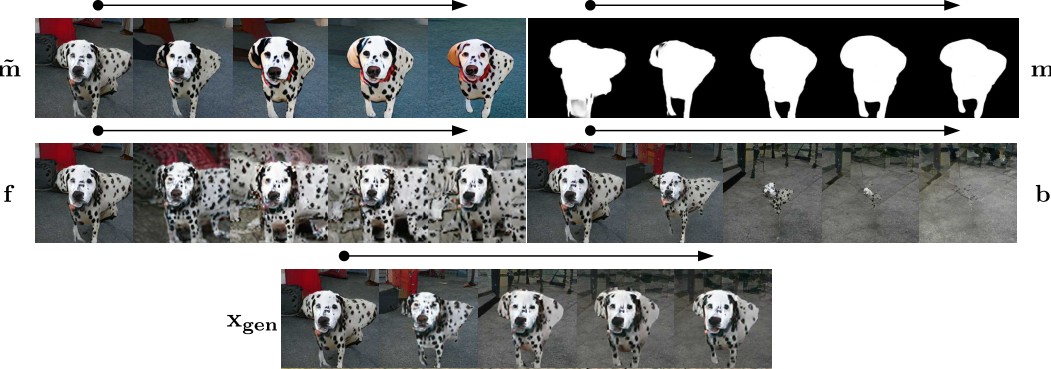

Figure 25: **IM Outputs over Training for 'dalmatian'** The arrows indicate the beginning and end of the training.

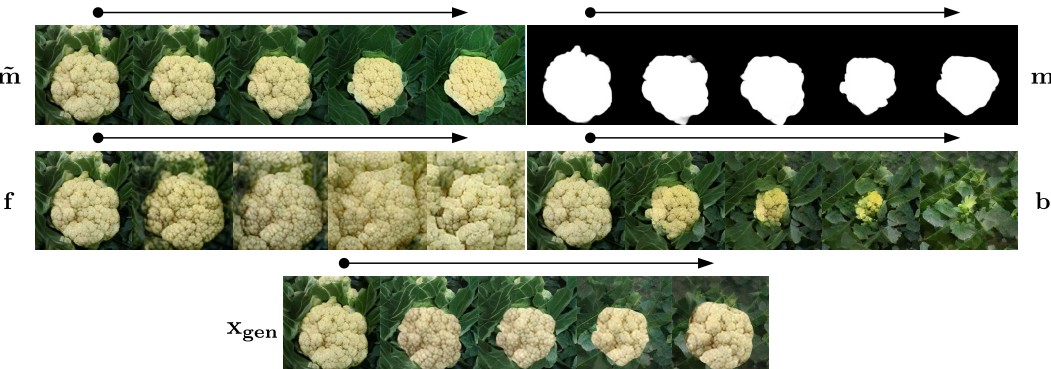

Figure 26: **IM Outputs over Training for 'cauliflower'** The arrows indicate the beginning and end of the training.

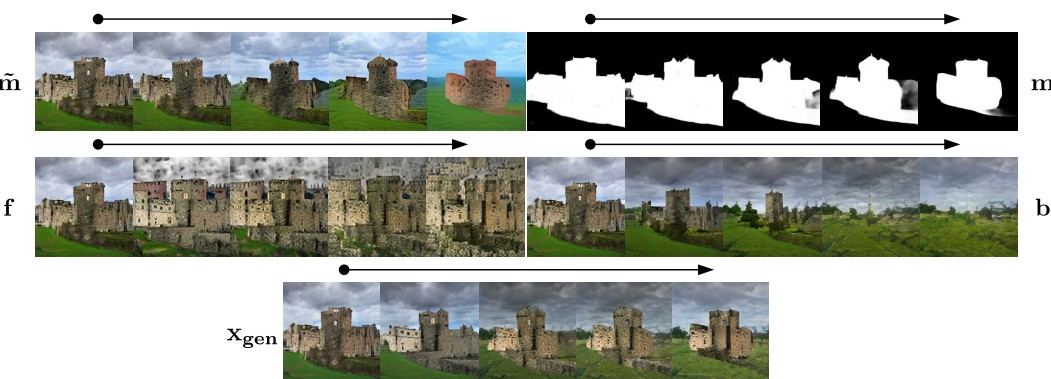

Figure 27: **IM Outputs over Training for 'castle'** The arrows indicate the beginning and end of the training.

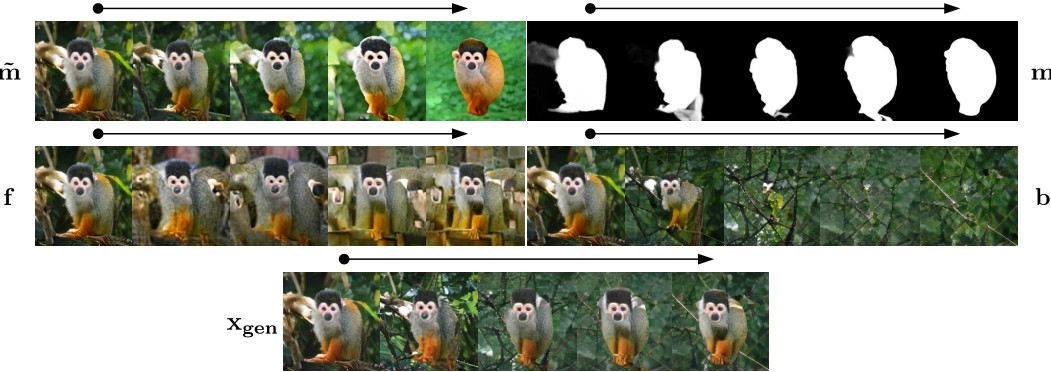

Figure 28: **IM Outputs over Training for 'ringtailed lemur'** The arrows indicate the beginning and end of the training.

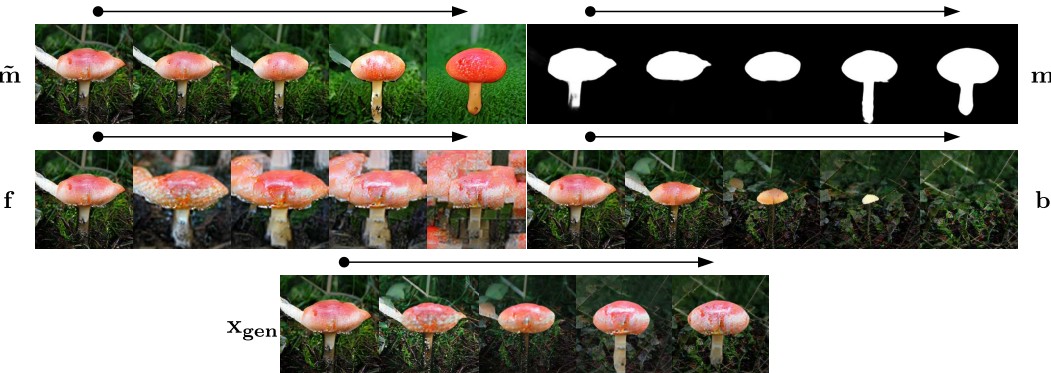

Figure 29: **IM Outputs over Training for 'mushroom'** The arrows indicate the beginning and end of the training.

## APPENDIX F  FAILURE CASES OF CGN

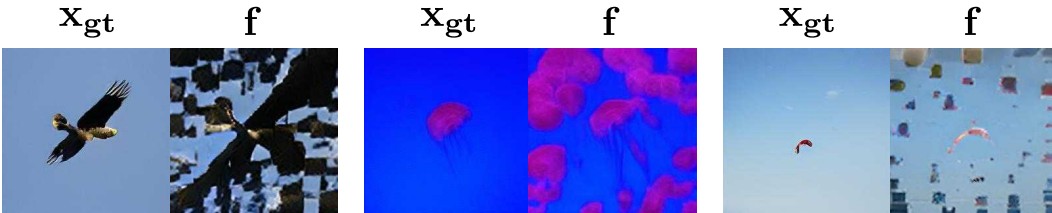

Figure 30: **Texture-Background Entanglement.** For relatively small objects, the texture maps can still show traces of the background. A possible remedy would be to choose the patch size for $\mathcal{L}_{text}$ dependent on the relative object size.

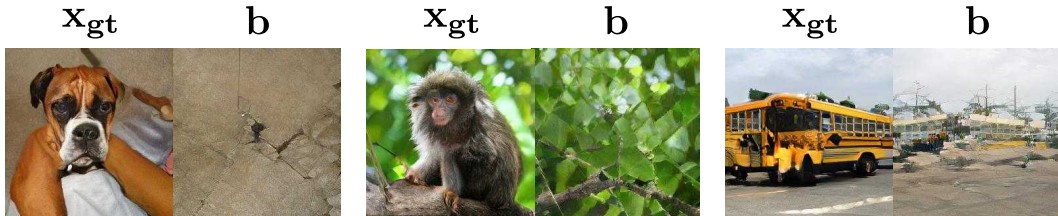

Figure 31: **Background Residues.** Especially for large objects, i.e., where large regions need to be in-painted, there can be faint artifacts visible. For the composite images, this is not a problem as an object will cover the residue.

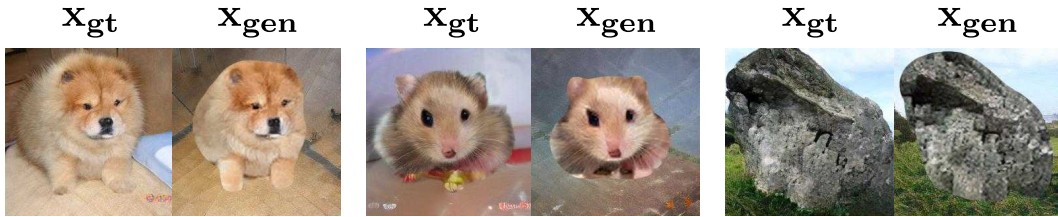

Figure 32: **Reduced Realism.** As evidence by the lower IS, the generated images $x_{gen}$ are generally lower in realism. This reduced realism is due to the constraints that we enforce and the simplified composition mechanism. A solution might be to add a shallow refinement network after the composer.

## APPENDIX G    COLLAPSING MASKS

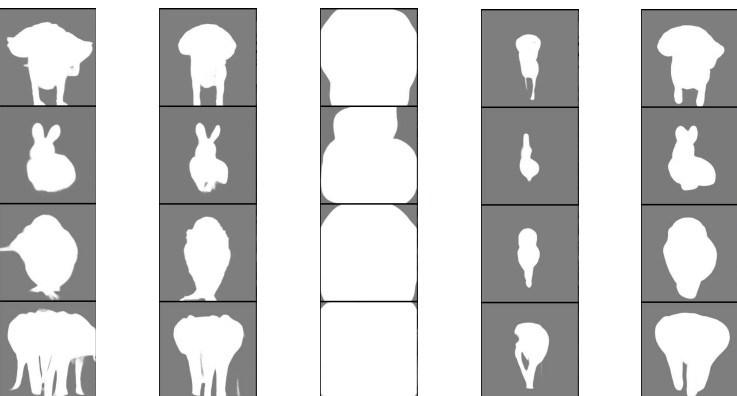

Figure 33: **Masks when disabling different losses.** From left to right: the beginning of training, training without $\mathcal{L}_{shape}$, training without $\mathcal{L}_{text}$, training without $\mathcal{L}_{bg}$, training with all losses. The third and fourth columns show the collapse of the masks, as described in section 4.2.

## APPENDIX H    CGN AUGMENTATION WITH IRM ASSUMPTIONS

We can drop the assumption of a priori knowledge of the causal signal and follow the same assumption as IRM: several environments with varying correlations, an invariant signal is considered causal. In the following, we train a CGN on double-colored MNIST. We then generate counterfactual data to train three classifiers:

- Shape classifier (SC): invariant wrt. object color and background color
- Object color classifier (OCC): invariant wrt. object shape and background color
- Background color classifier (BCC): one invariant wrt. object shape and object color

We can measure their respective performance in the environments used to train IRM. In these environments, only the shape is stably correlated with the label; the foreground and background color vary in their degree of correlation. Based on the results in Table 5, we can determine the shape to be the causal signal as only the test accuracy of SC is stable across environments.

| Degree of Correlation | Accuracy SC [%] | Accuracy OCC [%] | Accuracy BCC [%] |
|:---:|:---:|:---:|:---:|
| 90 % | $85.05 \pm 0.12$ | $58.72 \pm 0.67$ | $82.47 \pm 0.13$ |
| 95 % | $85.08 \pm 0.07$ | $62.69 \pm 0.58$ | $87.96 \pm 0.12$ |
| 100 % | $85.14 \pm 0.10$ | $65.05 \pm 1.32$ | $90.16 \pm 0.13$ |

Table 5: **Test Accuracy on double-colored MNIST.** We report the mean and standard deviation over three random seeds.

