# OpenReview forum: "Counterfactual Generative Networks"
_ICLR.cc/2021/Conference — ICLR 2021 Poster_

### Official Review · AnonReviewer4 · 2020-10-23
**Interesting approach but the presentation needs improvement**

**Rating:** 5
**Confidence:** 3

**Review:**

Deep neural network brittleness can be attributed to their tendency to latch on to spurious correlations in the training dataset. The proposal in the paper is to learn to generate samples where these correlations can be eliminated. To this end, the authors, distill trained conditional big gan into a transformation with explicit modules to capture the shape, texture of the foreground object, and the background. The distilled network is called Counterfactual Generator Network (CGN). Thus, an image can be generated with a background of one class, the shape of another class, and the foreground texture of a different class. Then a classifier with multiple heads is learned where each head predicts a class based on only one of the factors among shape, texture, and background.

The proposed approach is motivated by the assumption of independent mechanisms where different modules of the causal data generating process are independent of each other. Once the decomposition of a training image into shape, texture, and background is obtained, any component can be swapped to generate counterfactual data.

Pros:
+ The solutions provided to extract object masks, background and texture are interesting and scale to Imagenet dataset.
+  Shows that augmenting the training dataset with the generated counterfactual images can help improve robustness.

Cons + Questions:
- The presentation of the paper can be improved. It is not always clear if the causal structure is assumed to known. In Sec 2.2 SCM is defined, but the SCM for the MNIST or Imagenet is not provided. Do all the nodes in the CGN share the same noise or exogenous variables?
- The proposed method appears to assume that the causal structure is known. In this, it assumes it is made up of three nodes shape, texture, and background, and thus can limit the counterfactual generation ability. Many semantic changes cannot be achieved as evidenced by the fact that the counterfactual images are not realistic.
- in the invariant MNIST classification task it appears that the results are based on the assumption that the invariant feature - shape is known apriori. In practice, this information is not available. IRM does not assume this knowledge, so it does not seem comparison with IRM is fair in this case.
- Some related work that seems to be missing [1][2]

[1] Kocaoglu, Murat, et al. "Causalgan: Learning causal implicit generative models with adversarial training." arXiv preprint arXiv:1709.02023 (2017).

[2] Kaushik, Divyansh, Eduard Hovy, and Zachary C. Lipton. "Learning the difference that makes a difference with counterfactually-augmented data." arXiv preprint arXiv:1909.12434 (2019).

---

> ### Author Response · Authors · 2020-11-19
> **Answer to Reviewer 4 (1/2)**
>
> Thank you for your review. We appreciate your assessment of our approach as interesting and scalable. We also appreciate your suggestions to improve the presentation of our work. We will address your concerns in the following.
>
> __Q1: It is not always clear if the causal structure is assumed to known. In Sec 2.2 SCM is defined, but the SCM for the MNIST or Imagenet is not provided. Do all the nodes in the CGN share the same noise or exogenous variables?__
>
> Yes, we assume knowledge of the causal structure in all our experiments. As you correctly remark, the causal structure of our generative model is as follows
>
> \begin{align*}
> \mathbf{M} &:= f_{shape}(Y_1, U_1)\\\\
> \mathbf{F} &:= f_{text}(Y_2, U_2)\\\\
> \mathbf{B} &:= f_{bg}(Y_3, U_3)\\\\
> \mathbf{X_{gen}} &:= C(\mathbf{M}, \mathbf{F}, \mathbf{B})
> \end{align*}
>
> where $U_j$ is the exogenous noise, $Y_j$ is the label, $X_{gen}$ is the generated image, and $f_j$ and $C$ are the independent mechanisms. The general structure is the same for all experiments, with a slight simplification for MNIST: instead of $f_{bg}$ we assume a second texture mechanism $f_{text,2}$. There is no need for a globally coherent background in the MNIST setting.
>
> For the MNIST experiments, the noise variables are independent both during training and inference. For Imagenet, we deploy a cGAN for supervision. Hence, the CGN needs to use the same latent space as the cGAN, and all mechanisms use the same noise variables as input. Strictly speaking, this violates the assumption of independent noise. We accept this violation for the sake of stable optimization. However, during inference, we can draw $u$ independently for each independent mechanism (IM), hence implementing a proper SCM. We include a section in the appendix explicitly formulating the assumed causal structures for all experimental settings.
>
> __Q2: The proposed method appears to assume that the causal structure is known. In this, it assumes it is made up of three nodes shape, texture, and background, and thus can limit the counterfactual generation ability. Many semantic changes cannot be achieved as evidenced by the fact that the counterfactual images are not realistic.__
>
> This is indeed a limitation of our method; we assume that the causal problem structure is known. We add a paragraph in the additional discussion section to highlight this issue. As we mention in our discussion with Reviewer 1, an exciting future research direction might be to leverage causal discovery methods to find said structure.
>
> Further, our method is not limited to future extensions. One might add more IMs to model more fine-grained semantic changes. Also, as evidenced by our experiments, the counterfactual images are realistic enough to significantly influence a classifier’s preference on the main classification task (Imagenet) while maintaining performance.

---

> > ### Author Response · Authors · 2020-11-19
> > **Answer to Reviewer 4 (2/2)**
> >
> > __Q3: in the invariant MNIST classification task it appears that the results are based on the assumption that the invariant feature - shape is known apriori. In practice, this information is not available. IRM does not assume this knowledge, so it does not seem comparison with IRM is fair in this case.__
> >
> > One might argue that there is a tradeoff between these assumptions -- IRM needs several environments to work, we need to know that shape is the causal signal. However, we can also drop the assumption of a priori knowledge of the causal signal and follow the same assumption as IRM: several environments with varying correlations, an invariant signal is considered causal. For example, we can train a CGN on double-colored MNIST. We then generate counterfactual data to train three classifiers on CGN data:
> > 1. Shape classifier (SC): invariant wrt. object color and background color
> > 2. Object color classifier (OCC): invariant wrt. object shape and background color
> > 3. Background color classifier (BCC): one invariant wrt. object shape and object color
> >
> > We can measure their performance in the environments used to train IRM. Only the shape is stably correlated with the label; the foreground and background color vary in their degree of correlation. For this rebuttal, we implemented this setup and ran experiments. We report the mean and standard deviation over three random seeds:
> >
> > | Degree of Correlation 	| Accuracy SC    	| Accuracy OCC  	| Accuracy BCC  	|
> > |-----------------------	|----------------	|---------------	|---------------	|
> > | 90 %                  	| 85.05 $\pm$  0.12 % 	| 58.72 $\pm$ 0.67 % 	| 82.47 $\pm$ 0.13 % 	|
> > | 95 %                  	| 85.08 $\pm$ 0.07 % 	| 62.69 $\pm$ 0.58 %	| 87.96 $\pm$ 0.12 %	|
> > | 100 %                 	| 85.14 $\pm$ 0.10 % 	| 65.05 $\pm$ 1.32 %	| 90.16 $\pm$ 0.13 % 	|
> >
> > Based on these results, we can determine the shape to be the causal signal as only the test accuracy of SC is stable across environments. Of course, a point can be made that IRM is more general than our approach, as it does not assume a known causal structure. We include these discussion points and results in the paper.
> >
> > __Q4: Some related work that seems to be missing [1][2]__
> >
> > We include [1] in our work; we extensively discuss the differences to this work in our answer to reviewer 2. Kaushik et al. [2] demonstrate the value of counterfactual data in natural language inference. It is an interesting application of related ideas in a different domain -- thank you for making us aware of this reference; we include it in the related work section.
> >
> > __References__
> >
> > [1] Kocaoglu, Murat, et al. "Causalgan: Learning causal implicit generative models with adversarial training.", 2017, https://arxiv.org/abs/1709.02023
> >
> > [2] Kaushik, Divyansh, Eduard Hovy, and Zachary C. Lipton. "Learning the difference that makes a difference with counterfactually-augmented data.", 2019, https://arxiv.org/abs/1909.12434

---

> ### Author Response · Authors · 2020-11-22
> **More Questions?**
>
> Thank you again for your review. We believe that our rebuttal could address all your questions and concerns, and we hope to have changed your initial assessment for the better.
>
> As the discussion phase is nearing its end, we wondered if you might still have any concerns that we could address.
>
> Thank you for your time.

---

### Official Review · AnonReviewer2 · 2020-10-29
**Contribution Disconnected from Narrative, Missing Key Reference**

**Rating:** 5
**Confidence:** 5

**Review:**

The main idea of the paper, i.e., using independent causal mechanisms to generate interventional images, has already been explored by Kocaoglu et al. in Causalgan: Learning causal implicit generative models with adversarial training, ICLR'18. Same as here, the authors there also "view image generation as a causal process" and "structure a generator network as a structural causal model (SCM)" and use a conditional gan to generate the image from the labels. The generation used here based on three variables, i.e., shape, texture and background seem to be a special case. Therefore, the authors should definitely cite this work.

My general remark is that there is very little causality in the approach. The causal structure that is used in the generation of data is not different than a conditional GAN. This makes the claims in the introduction very disconnected from the actual methodology and the experiments in my opinion.

" we can intervene on a subset of them and generate counterfactual images "-> What the authors call counterfactual images are actually interventional images from a causal point of view. Please consider changing "counterfactual" to "interventional" throughout the paper. This will help clarify the distinction between interventional and counterfactual layers in Pearl's hierarchy.

"From a causal perspective, we maximize the average causal effect (ACE) of one IM on the classifier’s decision, while minimizing the ACE of all other IMs."
Can you formalize this claim? This does not seem trivial.

Could you explain "alpha blending"? This step is not motivated well and seems specific to the used dataset.

Even though the ImageNet experiments look impressive, I believe the main factor for success is in the deterministic and manually defined composition mechanism. Furthermore, I believe this composition is doing most of the disentangling during training.

Foreground and background segmentation use an existing method U2-Net which is used to create masks, or values for the variables used in the graph.

The intuition on comparing with other methods is missing. For example, why do you think training a classifier on interventionally augmented data performs better than IRM? Shouldn't this depend on the number of environments and degree of correlation? These are not reported.

"We, therefore, follow an augmentation strategy, i.e., we augment ImageNet with additional counterfactual images."
How many samples are added to the original data? I believe the amount of augmentation relative to the original dataset size is important.


%% AFTER REBUTTAL %%
Thank you for all the updates.

I would like to thank the authors for their humility in the rebuttal and for clarifying the paper's contributions. Accordingly, I will increase my score. However, I still believe Section 3.1's contribution, and the follow-up of using this to improve classifier robustness, is useful only for a very specific type of data and it is hard to assess its value from a practical point of view.

The fact that the authors were able to showcase that such counterfactual data augmentation improves classification is, although expected, useful in itself. However, performance improvement is only evident in colored MNIST, relative to GAN augmentation. Furthermore, R4 points out the important issue that the relevant causal feature is assumed to be known in the experiments. This information is normally not available and must be inferred by the classifier. The additional experiments provided by the authors during the rebuttal are welcome but they should be in the main paper rather than the appendix since this is the main setting where spurious correlations create problems. I believe the experimental section should put more weight on this setting.

In light of all this, I will provide a borderline score leaning towards rejection.

I encourage the authors to expand section 3 to settings that do not restrict the images to have one foreground object and a single background.

---

> ### Author Response · Authors · 2020-11-19
> **Answer to Reviewer 2 (1/4)**
>
> Thank you for your review. We will address your concerns in the following.
>
> __Q1: The main idea of the paper, i.e., using independent causal mechanisms to generate interventional images, has already been explored by Kocaoglu et al. in Causalgan: Learning causal implicit generative models with adversarial training, ICLR'18. Same as here, the authors there also "view image generation as a causal process" and "structure a generator network as a structural causal model (SCM)" and use a conditional gan to generate the image from the labels. The generation used here based on three variables, i.e., shape, texture and background seem to be a special case. Therefore, the authors should definitely cite this work.__
>
> We are aware of the work by Kocaoglu et al.; however, given the limited space for the initial submission, we decided to discuss more closely related work such as [1, 2]. However, we agree that it is helpful and necessary to contrast your proposed reference. Indeed, Kocaoglu et al. propose to structure a generator as a structural causal model. Concretely, they utilize two separate generative models: (i) a "causal controller," a model trained to learn the label distribution, the labels are binary (mustache, young). (ii) a generator conditioned on the generated labels.
>
> There are several aspects in which the work of Kocaoglu et al. significantly differs from our work:
> 1. There are substantial differences on a conceptual level. As we state: "We follow the argument that rather than training a monolithic network to map from a latent space to images, the mapping should be decomposed into several functions" (Section 2.2, Page 3). In our case, this latent space comprises noise and labels. CausalGAN learns an SCM on label level, then maps to images using a monolithic network. We do not map from noise and labels to images -- rather, we view the image formation process as an SCM, where different mechanisms interact with each other. Kocaoglu et al. do not use independent causal mechanisms, nor do they claim to do so.
> 2. Structuring the generator as an SCM is only one of our several contributions. For instance, Kocaoglu et al. do not deploy their GAN to generate data to train invariant classifiers, an essential aspect of our work.
> 3. Kocaoglu et al. assume a data set with fine-grained labels, and they report results on CelebA. It would not be straightforward to scale to more complex domains, such as Imagenet, where these kinds of labels are not available or very hard to obtain.
> 4. Applying a CausalGAN on our proposed MNIST variants and Imagenet would result in trivial solutions. The causal controller can not observe any variation in the label correlations. Hence, it would only learn deterministic mappings, as all the relevant factors of variation are 100 % correlated, i.e., a zero is always red (colored MNIST) or a banana is always "banana-textured" (Imagenet).
>
> We include the reference and discussion points in the paper.

---

> > ### Author Response · Authors · 2020-11-19
> > **Answer to Reviewer 2 (2/4)**
> >
> > __Q2: very little causality in the approach, the causal structure is not different than a conditional GAN, the claims in the introduction are very disconnected from the actual methodology”__
> >
> > The causal structure in a conventional conditional GAN (cGAN) is as follows:
> >
> > $$
> > \mathbf{X_{gen}} := f(U, Y)
> > $$
> >
> > where $U$ is the exogenous noise, $Y$ is the label, and $\mathbf{X_{gen}}$ is the generated image. A cGAN is a monolithic network that maps from a latent space (including labels) to images,  The causal structure of our proposed CGN:
> >
> > \begin{align*}
> > \mathbf{M} &:= f_{shape}(Y_1, U_1)\\\\
> > \mathbf{F} &:= f_{text}(Y_2, U_2)\\\\
> > \mathbf{B} &:= f_{bg}(Y_3, U_3)\\\\
> > \mathbf{X_{gen}} &:= C(\mathbf{M}, \mathbf{F}, \mathbf{B})
> > \end{align*}
> >
> > where $f_j$ and C are independent mechanisms. In Section 2.1, "Problem Setting," we explicitly motivate why a VAE fails in the investigated experimental settings. We also state that "the same behavior can be observed for unconstrained  GANs  (Goodfellow  et  al., 2014), as a GAN also approximates the training distribution." (Appendix A.1, Page 13). An unconstrained cGAN that follows the causal structure of Eq. (1) does not generate counterfactual images.
> >
> > Our central claims in the introduction are the following:
> > 1. By exploiting concepts from causality, this paper links two previously distinct domains: disentangled generative models and robust classification.
> > 2. This allows us to scale our experiments beyond small toy datasets typically used in either domain.
> >
> > Claim 1: As shown above, the causal structure of a conventional conditional GAN and a CGN is, in fact, not the same. Further, in our work, we leverage concepts from causality literature (independent mechanisms, interventions, counterfactuals) to draw the conceptual connection between disentangled representation learning, invariant representations, and robust classification. Taking a causal perspective to draw connections and to gain additional insights is in line with other work, such as [3,4]. Lastly, with our experiments on training invariant classifiers, we demonstrate the validity of these proposed connections.
> >
> > Claim 2: We demonstrate the scalability with our experiments on Imagenet.
> >
> > __Q3: What the authors call counterfactual images are actually interventional images from a causal point of view. Please consider changing "counterfactual" to "interventional" throughout the paper. This will help clarify the distinction between interventional and counterfactual layers in Pearl's hierarchy.__
> >
> > As we describe “these images [...] are generated from the intervention distribution" (Section 2.2, Page4). When we only sample from this distribution, they are indeed interventional images. However, we do have full control over the image formation process and the exogenous noise variables. Hence, we can answer counterfactual questions concerning a specific image. For example, we can generate an image, fix the input noise, and resample the background label to answer the question: "how would this image look like with a different background?" Similar reasoning can be found in [2] and [5]. We add this distinction to the paper and state where images are interventional and where they are counterfactual.
> >
> > Your comment made us reconsider our experimental setup as we did not leverage this aspect of our model before. We conducted additional experiments for this rebuttal. On MNIST, we can generate the same digit shape with several colors/textures, i.e., several counterfactuals per digit shape. Our new results indicate that it is beneficial to generate more than one counterfactual image per shape. We observe improvements across all MNIST variants (colored MNIST: 92.8 %-> 95.1%, double-colored MNIST: 88.0% -> 89.0%, Wildlife MNIST: 84.9%-> 85.7%). Our intuition is that several counterfactuals provide a more stable signal of the non-spurious factor.  We include our study about these effects in the appendix.
> >
> > __Q4: "From a causal perspective, we maximize the average causal effect (ACE) of one IM on the classifier's decision, while minimizing the ACE of all other IMs." Can you formalize this claim? This does not seem trivial.__
> >
> > We drop this claim in our revised version as it does not add any further insights or value.

---

> > > ### Author Response · Authors · 2020-11-19
> > > **Answer to Reviewer 2 (3/4)**
> > >
> > > __Q5: Could you explain "alpha blending"? This step is not motivated well and seems specific to the used dataset.__
> > >
> > > Alpha blending is not specific to the dataset; rather, it is specific to the task of image generation. As we explain in the paper: "we build on common assumptions from compositional image synthesis (Yang et al., 2017) and deploy a simple image formation model." (Section 3.1, Page 4). Alpha blending  (or alpha compositing) is a standard process from computer graphics and used to render pictures in separate layers and then combine them into a single composite image.
> > >
> > > __Q6: Even though the ImageNet experiments look impressive, I believe the main factor for success is in the deterministic and manually defined composition mechanism. Furthermore, I believe this composition is doing most of the disentangling during training.__
> > >
> > > Indeed, the composition mechanism is crucial to achieving disentanglement; hence, we highlight this in the text as an inductive bias. We will refine our writing to emphasize its importance. However, as we demonstrate in our loss ablation study, each loss is carefully motivated, and omitting only a single loss leads to inferior results and a failure to disentangle.
> > >
> > > We like to highlight that we consider this a strength of our work - in some instances, independent mechanisms can be interpreted as a "physical" mechanism [6] -- such as alpha blending. Our results demonstrate that it is beneficial to include knowledge (or assumptions) about the image formation process into a generative model. We find that integrating these simple and generic assumptions is crucial to scale disentanglement to ImageNet.
> > >
> > > __Q7: The intuition on comparing with other methods is missing. For example, why do you think training a classifier on interventionally augmented data performs better than IRM? Shouldn't this depend on the number of environments and degree of correlation? These are not reported.__
> > >
> > > Our work focuses on a setting where the spurious signal is a strong predictor of the label; hence we assume a correlation strength of at least 90 % between signal and label. This assumption is in line with latest related work on visual bias [5,7], which generally assumes a strong correlation to be above 95 %. Generally, we train and evaluate CGN on 100% correlated environments; we relax this constraint for IRM. We agree that our assumptions about the setting should be more clearly communicated. We include the prior discussion in the main text.
> > >
> > > Further, we did report the number of environments and their correlation. We referred the reader to the appendix for the baseline implementations (Appendix C.4), where we list these requested details.
> > >
> > > As for our intuitions, we believe that the difference between environments might be hard to pick up for IRM, especially if the number of environments is low. We train in two environments, the same experimental setting as originally proposed by [8]. For this rebuttal, we run additional experiments. We find that we can further improve IRM's performance by adding more environments, as it might be expected. More environments are crucial to achieving good results, and we can considerably improve IRM results on both benchmarks by training on 5 environments (double-colored MNIST: 78.9, Wildlife MNIST: 76.83%). However, the reached accuracy is still significantly lower than for our approach.  One might consider indefinitely increasing the number of environments to improve IRMs performance further. However, continually increasing the number of environments is an unrealistic premise and only feasible in simulated environments. Also, we needed to add scheduling for the gradient norm penalty weight to the original IRM formulation and carefully adjust hyperparameters to achieve these good results. We add the new, improved baseline results to table 2.

---

> > > > ### Author Response · Authors · 2020-11-19
> > > > **Answer to Reviewer 2 (4/4)**
> > > >
> > > > __Q8: How many samples are added to the original data? I believe the amount of augmentation relative to the original dataset size is important.__
> > > >
> > > > We discuss this question in detail in our answer to Reviewer 1. In short, on MNIST, more counterfactual data is always better. As for Imagenet, we settled on a Counterfactual/Real ratio of 1 for all of our experiments. A ratio below 1 leads to inferior performance on both Imagenet and IN-9. For a ratio above 1, the training time for a single epoch increases. However, the network also learns faster on the counterfactual data so that these two effects keep a balance, but we also do not observe an improvement over a ratio of 1.
> > > >
> > > >
> > > > Thank you again for your review. We hope that we could adequately address all of your concerns and that you might reconsider your initial evaluation.
> > > >
> > > > __References__
> > > >
> > > > [1] Li, Singh, Ojha, Lee, "MixNMatch: Multifactor Disentanglement and Encoding for Conditional Image Generation", 2020, CVPR, https://arxiv.org/abs/1911.11758
> > > >
> > > > [2] Besserve, Mehrjou, Sun, Schölkopf, "Counterfactuals uncover the modular structure of deep generative models", 2019, ICLR, https://arxiv.org/abs/1812.03253
> > > >
> > > > [3] Schölkopf, Janzing, Peters, Sgouritsa, Zhang, Mooij, "On causal and anticausal learning", 2012, ICML, https://arxiv.org/abs/1206.6471
> > > >
> > > > [4] Zhang, Zhang, Li, "A Causal View on Robustness of Neural Networks", 2020, NeurIPS, https://arxiv.org/abs/2005.01095
> > > >
> > > > [5] Goyal, Feder, Shalit, Kim," Explaining classifiers with causal concept effect (CACE)", 2019, https://arxiv.org/abs/1907.07165
> > > >
> > > > [6]  Schölkopf, "Causality for machine learning", 2019, https://arxiv.org/abs/1911.10500
> > > >
> > > > [7] Wang, Qinami, Karakozis, Genova, Nair, Prem, Russakovsky, "Towards fairness in visual recognition: Effective strategies for bias mitigation", 2020, CVPR, https://arxiv.org/abs/1911.11834
> > > >
> > > > [8] Arjovsky, Bottou, Gulrajani, Lopez-Paz, "Invariant risk minimization", 2019, https://arxiv.org/abs/1907.02893

---

> > > > > ### Comment · AnonReviewer2 · 2020-11-20
> > > > > **Thank you for the detailed response**
> > > > >
> > > > > "We are aware of the work by Kocaoglu et al.; however, given the limited space for the initial submission, we decided to discuss more closely related work"
> > > > > Part of the claimed contribution significantly overlaps with the aforementioned paper and the authors chose to not cite it. I hope the authors will acknowledge that such intentional omittance may mislead the reviewers in wrongly evaluating the paper's contributions. I hope the authors will address this issue in the current and future versions of their manuscript. Given that the authors were aware of this work, I believe they should cite it in the introduction when they say "we can interpret the generation of an image as a causal process." instead of only in page 9.
> > > > >
> > > > > "Kocaoglu et al. do not use independent causal mechanisms, nor do they claim to do so."
> > > > > Independent mechanism refers to using an SCM: Each variable has a generating mechanism that is independently manipulable from the other generative mechanisms. Therefore, it uses independent causal mechanisms since it models an SCM.
> > > > >
> > > > > About the papers' contributions:
> > > > >
> > > > > "disentanglement of independent mechanisms"
> > > > > I still believe the narrative is disconnected from what is actually happening in the paper. The hand-coded composition mechanism is essentially forcing the network to learn what to output for each pre-specified task. This is different from and weaker than the existing work that attempts to disentangle independent mechanisms such as InfoGAN and its follow-ups which do not use any such supervision, since it is specialized to this very specific task. The mechanisms in this image generation are instead "hand-coded" to be independent.
> > > > >
> > > > > "We add this distinction to the paper and state where images are interventional and where they are counterfactual."
> > > > > "additional IRM experiments"
> > > > > I appreciate these updates.

---

> > > > > > ### Author Response · Authors · 2020-11-21
> > > > > > **Response**
> > > > > >
> > > > > > Thank you for your reply; we highly appreciate you taking the time to discuss with us.
> > > > > >
> > > > > > __Missing Reference to Kocaoglu et al.__
> > > > > >
> > > > > > As outlined above, we disagree with the amount of overlap to Kocaoglu et al.. Not one of the contributions we list in the introduction (linking disentanglement and robust classification; scaling to ImageNet; generating high-quality counterfactual images with direct control over shape, texture, and background; usefulness of the generated counterfactual images for downstream tasks; emerging properties of the generative model) is addressed by Kocaoglu et. al.
> > > > > >
> > > > > > However, we agree that including this reference in the initial submission would have been better. We hope the missing reference did not lead any reviewer to significantly misjudge our actual contributions. Further, we also do not intend to take ownership of the idea to interpret image generation as a causal process. Still, we see how it could be misinterpreted in the current version - we follow your suggestion and cite the related work in the respective passage in our latest version.
> > > > > >
> > > > > > __Disconnected Narrative.__
> > > > > >
> > > > > > Thank you for expanding on this point; we do understand your concerns more clearly now. We think the problem with our current version is two-fold, and we aim to address both aspects.
> > > > > > 1. __The strength of the composition mechanism.__ First, we like to highlight again that a foreground/background assumption is generic and not necessarily restricted to the task of image classification. That being said, it remains to be seen if the idea of structuring a generator into and learning these mechanisms scales to other problems where we cannot apply a foreground/background assumption, e.g., medical images (as discussed in our answer to Reviewer 1). Here, we can not rely on the powerful bias of the defined composition. However, there may be equally useful biases in other domains that enable us to learn the SCM. Yet, this investigation is beyond the scope of our work. Therefore, we add a disclaimer to the discussion section: “The composition mechanism is a powerful bias, and crucial to making our model work. In other domains, equally strong biases may need to be identified to enable learning the SCM.”
> > > > > > 2. __Comparison to other disentanglement methods.__ In the experiments section, we repeatedly write, “our approach disentangles the independent mechanisms.” We understand now how this formulation can be misleading. We define the disentangled structure and get the respective subnetworks to learn inside this structure; hence, the approach does not disentangle the signals by itself. This aspect is, in fact, weaker than the more general Info-GAN and its follow-up work. To resolve this disconnect, we propose the following changes:
> > > > > > - An additional disclaimer in the discussion section contrasting to standard disentanglement works:
> > > > > > “ [...] we assume the causal structure to be known. This assumption is substantially stronger than the ones in more general standard disentanglement frameworks (Chen et al. 2016, Higgins et al., 2017).”
> > > > > >
> > > > > > - “ Does our approach reliably disentangle IMs on datasets of different complexity?” (Section 4, Page 6)
> > > > > > &rightarrow; “Does our approach reliably learn the disentangled IMs on datasets of different complexity?”
> > > > > >
> > > > > > - “Does our approach disentangle the independent mechanisms?” (Section 4.1 heading, Page 6)
> > > > > > &rightarrow; “Does our approach learn the disentangled independent mechanisms?”
> > > > > >
> > > > > > - “The CGN successfully disentangles shape, texture, and background…” (Figure 4 caption, Page 7)
> > > > > > &rightarrow; “The CGN successfully learns the disentangled shape, texture, and background mechanisms…”
> > > > > >
> > > > > > We hope our proposed changes resolve the issue of a disconnected narrative, and we do believe that these changes remove statements that could be perceived as overclaims. If not, we are also open to further suggestions. Thank you again for your time and input.

---

### Official Review · AnonReviewer3 · 2020-11-03
**Official Blind Review #3**

**Rating:** 7
**Confidence:** 3

**Review:**

--- Summary ---

This paper proposes a new generative model that generate images from 3 seperate aspects: foreground masks (shapes), forground texture, and backgrounds. Then they convexly mix these 3 aspects into one image. By doing so, they can vary each aspect individually without changing other aspects, enabling the model to generate counterfactual images. For example, we can generate a cat shape with telephene texture and sea background, which would not exist in natural images. They show that in several colorful MNISTs datasets their methods can generate new combinations of images. In ImageNet, by using the pre-trained BigNet GAN as backbones and pretrained U-Net for foreground object masks, they can distill the knowledge in BigNet into these 3 seperate aspects and generate counterfactual natural images.

--- Pros ---

1. Intersting ideas of combining pre-trained models with novel loss function that helps disentangle these 3 aspects.
2. Plausible and intriguing counterfactual examples.
3. Strong improvements in colorful MNIST datasets.

--- Major comments ---

1. The claim that "we are able to reduce the gap while achieving hig accuracy on IN-9" is not true. In Table 4, the method IN+CGN has lower accuracy than IN alone, and it reduces the gap by lowering the original performance in Mixed-same. So it does not actually improve the performance.

2. The above makes me wonder the ImageNet experiment does not actually generate plausible enough images that help improve accuracy, probably due to its unnatural image generation as evidenced by relatively low IS scores (130).
Especially seeing there are 6 lambdas to tune listed in Appendix B.3 and other hyperparameters like learning rates etc, this method might not be very practical in high-dimensional natural images. Maybe authors can be honest about it. Or illustrate what's the best way to tune this method, what has been attempted etc.

3. The current setting seems to a bit limited that requires a single foreground and background. For example, designing independent mechanisms for medical imaging might not be easy with no exact foreground/background boundaries.

4. In Table 3, maybe authors can evaluate on some difficult ImageNet datasets like ImageNet-A to assess if the performance improves.

--- Minor comments ---

1. The loss descriptions in Appendix B should be moved to main text to help readers understand the method. Details like how to pick $\tau$ for shape loss should be mentioned.

2. The name "pre-masks" is confusing that originally I think it's a binary mask, but instead it's a colorful image.

3. More failure examples in Appendix E will better help readers understand its limitations.

4. The figure 5 and the Appendix D should also include the final image that combines m, f and b to better assess its improvement over the training stages. Also, having an arrow in Figure 5 or specifying epochs might better help readers understand it is showing the transitions.

5. The caption in Figure 24 is wrong: should be columns instead of rows.

--- Evaluations ---

Overall I like this paper. The method is interesting to read and the examples are interesting. The experiments are thorough and do show some improvements in colorful MNISTs. The method, unfortunately, does not seem to work that well in ImageNet, and does not improve generalization performance. I encourage the authors to be upfront about the limitations of this method and write better descriptions of loss and hyperparameters tuning.

---

> ### Author Response · Authors · 2020-11-19
> **Answer to Reviewer 3 (1/2)**
>
> Thank you for your review. We appreciate your assessment of our generated counterfactual images as "plausible and intriguing" and our experiments as "thorough," and you highlighting our idea of combining pre-trained models with novel loss functions. We address your raised concerns in the following, starting with your primary concerns.
>
> __Q1: IN-9 results - IN+CGN has lower accuracy than IN alone, and it reduces the gap by lowering the original performance in Mixed-same. So it does not actually improve the performance. The above makes me wonder the ImageNet experiment does not actually generate plausible enough images that help improve accuracy, probably due to its unnatural image generation as evidenced by relatively low IS scores (130).__
>
> We conducted additional experiments for this rebuttal. Our updated results show a slight improvement on Mixed-Rand (IN: 78.9 %, IN+CGN: 80.1%). We achieve this improvement by increasing the overall batch size (both real and counterfactual batches) during training. However, we continue to observe a drop on Mixed-Same (IN: 86.2 %, IN+CGN: 83.4%). Your question still raises a valid point - why does the performance on Mixed-Same decline?
>
> We hypothesize that there is an additional domain shift that our approach does not address. A model trained on IN achieves the following accuracies: IN-9 (original): 95.6%, Mixed-Same: 86.2%, Mixed-Rand: 78.9%. A stated in [1], Mixed-Same controls for artifacts from image processing. These artifacts lead to a drop in performance of 9.4% - even though the background is still 100% correlated with the label. Hence, without handling these artifacts, 86.2% appears to be the upper boundary if all available factors of variation are used for classification -- including the background. It seems natural that a classifier which uses the background signal less and does not address these artifacts would further drop in performance, i.e., below 86.2%
>
> We also believe that your argument of unnatural images is correct. As we remark in our answer to Reviewer 1, we believe that the quality of the synthetic images is not high enough yet to *improve* performance on ImageNet. On the one hand, the constraints we enforce on our model result in less realistic images, as evidenced by the lower IS and the failure cases we present (see below). On the other hand, even state-of-the-art generative models (with higher IS) are not good enough yet to generate data for training competitive ImageNet classifiers, see [2].
>
> __Q2: Especially seeing there are 6 lambdas to tune listed in Appendix B.3 and other hyperparameters like learning rates etc, this method might not be very practical in high-dimensional natural images. Maybe authors can be honest about it. Or illustrate what's the best way to tune this method, what has been attempted etc.__
>
> Thank you for bringing up this point. We agree our initial submission was lacking details about how to find good hyperparameters. As we previously described, during training, we measure the Inception score (IS) and the mean value of the masks $\mu_{mask}$ to detect mask collapse. We also observe the generated images for a fixed noise vector; see the outputs in Figure 5. Our overall objective is a high IS and a stable $\mu_{mask}$. Further, we aim for high-quality output of all IMs (Masks: binary, capture only class-specific parts; Textures: no background/global shape visible, Background: no trace of foreground objects visible). We observe these outputs for several classes during optimization. As we show in our loss ablation study, all losses are necessary for disentanglement. However, in our experience, the hyperparameters can be tuned mostly independently from each other, i.e., a better lambda for the mask loss does not influence the quality of the texture maps much. We include this description in the appendix.
>
> __Q3: limited setting (requires a single foreground and background). Designing independent mechanisms for medical imaging might not be easy with no exact foreground/background boundaries.__
>
> The setting of a single foreground and background is indeed a limitation of our current work. Your suggested example is an exciting one. A possibility would be to avoid the foreground/background assumption and train IMs to generate domain-specific object instances. For example, for brain scans, different IMs could model different brain regions that can only be composed in a certain way. The composition function can be informed by domain-specific knowledge. We include a discussion of the current limitations regarding the points above in the discussion section.

---

> > ### Author Response · Authors · 2020-11-19
> > **Answer to Reviewer 3 (2/2)**
> >
> > __Q4: In Table 3, maybe authors can evaluate on some difficult ImageNet datasets like ImageNet-A to assess if the performance improves.__
> >
> > As suggested, we evaluate our invariant classifier ensemble on Imagenet-A. We observe a marginal improvement of the Top-1 accuracy from 0 % to 2.6 %, which is in the same ballpark as augmentation with Stylized Imagenet (2.3 %). As we discussed above, this only minor improvement might be due to the remaining reality gap between synthetic and real images.
> >
> > __Q5: Details like how to pick lambda for shape loss should be mentioned.__
> >
> > Tau has a direct interpretation: if set to 0.1, the expected mean of the mask should be in the interval of [0.1, 0.9] -- the main object should occupy more than 10% and less than 90% of the image. We believe that this a reasonable general assumption and will be adding our explanation to the paper. Ablating this value might improve the IS value, but we expect only minor improvements; hence, we did not investigate it further.
> >
> > __Q6: More failure examples in Appendix E will better help readers understand its limitations.__
> >
> > We add additional illustrations of the described failure cases to give a more detailed impression of these cases. Further, we also highlight an additional failure case: a decrease in realism due to the optimization constraints and the simplified composition.
> >
> > __Q7: The figure 5 and the Appendix D should also include the final image that combines m, f and b to better assess its improvement over the training stages. Also, having an arrow in Figure 5 or specifying epochs might better help readers understand it is showing the transitions.__
> >
> > We added the composite image and the suggested arrows in the mentioned figures. We also include the composite images in Appendix C, where we show the individual IM outputs. We agree that this will improve the reader's understanding of the training process.
> >
> > Thank you again for your comprehensive review (including the appendix) and your valuable suggestions!
> >
> > __References__
> >
> > [1] Xiao., Engstrom, Ilyas, and Madry, "Noise or signal: The role of image backgrounds in object recognition", 2020, https://arxiv.org/abs/2006.09994
> > [2] Suman, Vinyals, "Seeing is not necessarily believing: Limitations of biggans for data augmentation", 2019, ICLR Workshop LLD, https://openreview.net/forum?id=rJMw747l_4

---

> > > ### Comment · AnonReviewer3 · 2020-11-20
> > > **Thank you for the rebuttal**
> > >
> > > First, thank you for the thorough updates - I looked again for the revision and the paper improves quite a bit!
> > >
> > > Q1: Although it's good to see this method at least improves some (1.1%) in Mixed-Rand, it does not seem to be fair because this new change (enlarge the batch size) should also be applied to your baselines. But I don't see numbers updated in Table for your baselines.
> > >
> > > Q2-Q7: thanks for clarifying.
> > >
> > > Additional questions as I read through the updated pdfs:
> > > (1) In Methods section, "It is also possible to train on interventional images xIV . Empirically, we find that counterfactual images improve performance over interventional ones. We hypothesize that counterfactuals provide a more stable signal."
> > > This is unclear what you mean internentional images here and what's difference to CFs.
> > >
> > > (2) In appendix ablation study, "We find that the higher the ratio, the better." It's simply not true as shown in Figure 7 that some mid-ratios are best in Colored MNIST and Double-colored MNIST. In wildlife MNIST yes ratio=20 is the best but ratio=10 gets worse than 5 or 1.
> > >
> > > (3) For R2's (valid) concern, now the IRM baselines of how to choose environments for MNISTs are mentioned, but you should put in the main text instead of Appendix C.4. Specifically, the sentence:
> > > "We then train IRM on 2 environments (90% and 100% correlation) or 5 environments (90%, 92.5%, 95%, 97.5%,
> > > and 100% correlation)."

---

> > > > ### Author Response · Authors · 2020-11-21
> > > > **Response**
> > > >
> > > > Thank you for taking the time to continue discussing with us; we highly appreciate it!
> > > >
> > > > __Q1. Increasing batch size for the baselines__
> > > >
> > > > We believe the reason for the improvement is a better gradient for the counterfactual signal. For the baselines, we report the performance of pre-trained networks (IN: torchvision.models.resnet50, SIN: https://github.com/rgeirhos/texture-vs-shape, Mixed-Rand: https://github.com/MadryLab/backgrounds_challenge). These networks were selected based on their performance on the ImageNet validation set.
> > > > Only the SIN and IN+SIN baselines may profit from a larger batch size, as the gradient for the shape in the stylized images might be better. We do not expect the performance to improve, as the background signal can still be fully utilized. This phenomenon is demonstrated in the SIN baseline, where the BG gap even increases. However, we will still run this experiment and update the results if they improve in favor of the baseline (this update might be after the discussion deadline).
> > > >
> > > > __Additional Questions__
> > > > 1. We add a sentence to contrast these two options in the cited passage more clearly.
> > > > 2. This is, in fact, a mistake. It should not say “the higher the ratio, the better”; instead, “the higher the number of counterfactual data points, the better” -- this is true in all cases. Thank you for catching this one.
> > > > 3. We agree. We add your suggestion to the main text.
> > > >
> > > > Thank you again for your time and input.

---

### Official Review · AnonReviewer1 · 2020-11-09
**Interesting conceptual connection between causality, disentangled representation learning, invariant representations and robust classification**

**Rating:** 8
**Confidence:** 4

**Review:**

This paper presents an interesting conceptual advance connecting causality, disentangled representation learning, invariant representations and robust classification.
The authors propose a Counterfactual Generative Network (CGN), which is basically "modular" generative adversarial network that can independently control the generation of independent factors of variations in the data corresponding to Independent Mechanism, i.e. independent factors in the structural causal model of the data. In the context of generating natural images like those comprising the ImageNet dataset, the CGN once trained can be used to generate high-quality counterfactual images with direct control over of factors of variations determining the content of an image shape, texture, and background. These generated samples obtained by independently and uniformly sampling over factors of variation can be used to train a classifier to achieve out-of-domain robustness. The authors show indeed show in simulation that this procedure works as a data augmentation procedure that increases out-of-domain robustness while only marginally degrades the overall accuracy. As the authors explain, this can be thought of as a generalization of "domain randomization".
Additionally, CGN can be used as a generative model of high-quality binary object masks and unsupervised image inpainting.
The authors also carry out extensive ablation studies that quantify the contribution of the different training costs for CGN to the overall quality (measures as Inception Score) of the generated counterfactual images.
This is first and foremost an "idea paper" putting forth a very interesting conceptual proposal. This is then empirically validated on out-of-distribution classification tasks in different versions of colored MNIST, and a coarse grained subsed of ImageNet.
A natural question for the authors is whether and by how much this type of data augmentation might help in realistic large-scale classification scenarios. Another natural question would be to quantify the effect of the counterfactual data augmentation in terms of trade-off between performance on in-distribution and out-of-distribution samples.
Lastly, it would be interesting to know whether the authors have any thought on how to generalize their architecture to other setting, i.e. how to isolate Independent Mechanisms in a general domain and what type of domain knowledge is necessary to do that effectively.

---

> ### Author Response · Authors · 2020-11-19
> **Answer to Reviewer 1**
>
> Thank you for your review. We appreciate your assessment of our work as an “interesting conceptual advance.” We, too, believe that our main contribution is connecting the fields of causality, disentanglement, and robust classification. With further advances in generative modeling and synthetic-to-real literature, we are optimistic that we can further improve results on realistic datasets like Imagenet. In the following, we like to address all the questions you raised.
>
> __Q1: whether and by how much this type of data augmentation might help in realistic large-scale classification scenarios.__
>
> As we show in Table 3 (Shape vs. Texture), our invariant classifier ensemble is able to trade off the different factors of variation on the full Imagenet classification task. Interestingly, we can strongly influence the classifier’s preferences while maintaining good performance on the (large-scale) classification task  - even more than augmenting with Stylized Imagenet.
>
> Our intuition is that the quality of the synthetic images is not high enough yet to *improve* performance on ImageNet itself. On the one hand, the constraints we enforce on our model result in less realistic images, as evidenced by the lower IS. On the other hand, even state-of-the-art generative models (with higher IS) are not good enough yet to generate data for training competitive ImageNet classifiers, see [1].
>
> __Q2: Effect of the counterfactual data augmentation in terms of trade-off between performance on in-distribution and out-of-distribution samples.__
>
> This question is indeed interesting, and we ran additional experiments both on MNIST and Imagenet. We did not observe a difference in our initial experiments when training on both OOD and IID data, or only on OOD data. We increase the OOD/IID ratio beyond 1 (the initially used ratio) in our new experiments. Interestingly, on MNIST, there seems to be no downgrade in performance when we increase the ratio of OOD/IID samples - more OOD data always improves the performance on the test set. This effect holds up to a ratio of 20; beyond that, we see stagnation. By increasing this ratio, we achieve a substantial improvement over our initial reported numbers (Colored MNIST 86.4 % -> 95.1%, Double-Colored MNIST 85.3% -> 89.0%, Wildlife MNIST 80.5% -> 85.7%). These results illustrate the potential of generated counterfactual data, especially in less complex domains. We add these new results to the paper.
>
> As for Imagenet, we settled on a ratio of 1 for all of our experiments, based on similar results by [1]. A ratio below 1 leads to inferior performance on both Imagenet and IN-9. For a ratio above 1, the training time for a single epoch increases. However, the network also learns faster on the counterfactual data so that these two effects keep a balance, but we also do not observe an improvement over a ratio of 1.
>
> __Q3: Lastly, it would be interesting to know whether the authors have any thought on how to generalize their architecture to other setting, i.e. how to isolate Independent Mechanisms in a general domain and what type of domain knowledge is necessary to do that effectively.__
>
> We, too, are very interested in how to generalize our ideas to different settings. A straightforward way to extend our proposed CGN is to add several branches for object instances. This type of disentanglement might produce useful training data for semantic (instance) segmentation. Another possible avenue is to build on advances in 3d generative modeling [2, 3] to generate data to train classifiers invariant to object pose.
>
> It is generally hard to state what domain knowledge is necessary and may depend heavily on the respective domain. With our focus on image classification, we incorporated general information about compositional image formation, i.e., alpha blending. That being said, it would be interesting to explore ideas from causal discovery to isolate IMs in a domain-agnostic manner, e.g., via meta-learning [4].
>
> __References__
>
> [1] Suman, Vinyals, “Seeing is not necessarily believing: Limitations of biggans for data augmentation”, 2019, ICLR Workshop LLD, https://openreview.net/forum?id=rJMw747l_4
>
> [2] Nguyen-Phuoc, Li, Theis, Richardt, Yang, “Hologan: Unsupervised learning of 3d representations from natural images”, 2019, ICCV, https://arxiv.org/abs/1904.01326
>
> [3] Liao, Schwarz, Mescheder, Geiger, “Towards unsupervised learning of generative models for 3d controllable image synthesis”, 2020, CVPR, https://arxiv.org/abs/1912.05237
>
> [4] Bengio, Deleu, Rahaman, Ke, Lachapelle, Bilaniuk, Goyal, Pal “A meta-transfer objective for learning to disentangle causal mechanisms”, 2019, ICLR, https://arxiv.org/abs/1901.10912

---

### Author Response · Authors · 2020-11-19
**Update Summary**

We like to thank all reviewers again for their valuable feedback regarding theory, experiments, and presentation. We highlight the revised text in red. We added the following updates to our initial submission.

--- 19 Nov 2020 ---

**Major Updates**
- Include suggested relevant work (Kocaoglu et al., Kaushik et al.) and discussion thereof
- Explicitly formulated the causal structure for all experiments
- Added clear distinction between counterfactual and interventional images
- We added a dedicated discussion section addressing several points raised by the reviewers (limitations, realism, generality, future extensions)
- More examples and discussion of failure cases
- Study of CGN augmentation over several environments with varying correlation
- Ablation study over OOD/IID ratio and number of counterfactual samples (for MNIST experiments)
- improved results and description for IRM baseline
- Explicit formulation of the correlation strength we consider in our experiments
**Minor Updates**
- Added composite images and arrows in Figure 5 and Appendix C.1 and D.
- We show more examples of different class types in Appendix C.1.
- Details description of how to find good hyperparameters
- Justification of our choice for $\tau=0.1$ for the shape loss
- Improved results on IN-9
- Emphasis on the importance of the composer
- Drop of claim: "From a causal perspective, we maximize the average causal effect (ACE) of one IM on the classifier's decision, while minimizing the ACE of all other IMs."
- Typo in the caption of Figure 24

--- 21 Nov 2020 ---
- Moving references to introduction to contrast our own contributions more clearly
- Additional discussion of the composition mechanism in the discussion section
- Disclaimer about the strength of our assumptions in the discussion section
- Reformulations in the experiments section to contrast to general disentanglement frameworks
- Improved explanation of interventional images
- Fixed mistake in MNIST ablation study (Appendix A.3)
- Move the description of IRM environments to the main text

---

### Decision · Program_Chairs · 2021-01-07
**Final Decision**

**Decision:**

Accept (Poster)

**Comment:**

The paper presents a "conceptual  advance connecting causality, disentangled representation learning, invariant representations and robust classification". Authors propose to decompose the image generation process to independent mechanism that can be composed (foreground masks (shapes), forground texture, and backgrounds), allowing for a specific image to generate counterfactuals , by changing some variations factors, while keeping other fixed. One can use interventional data to augment classifiers, this can lead in certain cases to improvement in accuracy and in other in improving the robustness.

There was concerns about the clarity of the paper regarding the structured causal model considered and its applicability beyond image generation, experimental protocol for choosing hyperparameters (loss scaling and ratios of real data and interventional samples ) and some missing references. The rebuttal of the authors and their updated paper reflected comprehensively all those concerns and addressed them, highlighting limitations of the method and adding more examples of its failures.

I liked the ideas and concepts in  this paper , and it will be exciting to generalize such generative approach to other domains, this  work is a first step. I think it will be good addition to ICLR program